# CATArena: Evaluation of LLM Agents Through Iterative Tournament Competitions

## Abstract

Large Language Model (LLM) agents have evolved from basic text generation to autonomously completing complex tasks through interaction with external tools. However, current benchmarks mainly assess end-to-end performance in fixed scenarios, restricting evaluation to specific skills and suffering from score saturation and growing dependence on expert annotation as agent capabilities improve. In this work, we emphasize the importance of learning ability, including both self-improvement and peer-learning, as a core driver for agent evolution toward human-level intelligence. We propose an iterative, competitive peer-learning framework, which allows agents to refine and optimize their strategies through repeated interactions and feedback, thereby systematically evaluating their learning capabilities. To address the score saturation issue in current benchmarks, we introduce CATArena, a tournament-style evaluation platform featuring four diverse board and card games with open-ended scoring. By providing tasks without explicit upper score limits, CATArena enables continuous and dynamic evaluation of rapidly advancing agent capabilities. Experimental results and analyses involving both minimal and commercial code agents demonstrate that CATArena provides reliable, stable, and scalable benchmarking for core agent abilities, particularly learning ability and strategy coding.

## 1 Introduction

With the rapid evolution of agents powered by large language models (LLMs), their capabilities have far surpassed simple text generation. By actively invoking external tools, LLM agents have significantly expanded the boundaries of artificial general intelligence (AGI). These agents are now able to autonomously complete complex, multi-step tasks that are previously considered beyond their reach, such as developing software project (Manish, 2024; Hu et al., 2025b), intelligently performing strategic planning (Belle et al., 2025), and learning user preference (Gao et al., 2024).

Existing benchmarks mainly focus on end-to-end performance in specific tasks, such as code generation (Yang et al., 2024), AI research (Nathani et al., 2025), and GUI automation (Wang et al., 2024). These benchmarks provide detailed observations and analyses of LLM agents' abilities within particular scenarios and have driven significant progress in the field. However, there are important limitations to these approaches. First, *the scores obtained in these end-to-end benchmarks only reflect performance on specific tasks*, whereas an agent's overall capability is composed of multiple fundamental skills working together. Second, *the absolute scores in these benchmarks, which are typically based on objective correctness, have an upper bound*. As agents become increasingly powerful, maintaining and updating these benchmarks requires additional expert-level annotation, and the level of required expertise continues to rise. In light of these challenges, there is an urgent need for a quantifiable and continuously evolving benchmark that systematically measures and analyzes the fundamental sub-abilities of agents.

Previous research has shown that self-learning is an essential ability for agents to achieve human-level intelligence (Gao et al., 2025; Zhu et al., 2025). Beyond self-learning, agents, similar to humans, also engage in peer learning, which enables collective evolution through interactions and shared experiences (Liu et al., 2024). During this evolutionary process, agents receive feedback from their environment and continually improve themselves. This capacity for learning and adaptation is indispensable for LLM agents, as it prepares them for ongoing evolution and more complex

challenges. To systematically evaluate this crucial ability, we propose an iterative peer-learning-based competitive framework for LLM agents. In each iteration, agents are required to revise and update their strategies based on the outcomes and policies observed in previous rounds of competition. This process exemplifies in-context learning, where agents leverage contextual information to adapt and evolve without changing their underlying model parameters. After every update, the agent policy codes are executed and competed against each other, generating dynamic performance rankings. Through this peer-learning architecture, we gain valuable insights into the learning abilities of LLM agents.

Building on this peer-learning framework, we introduce CATArena (**C**ode **A**gent **T**ournament **Arena**), which utilizes four open-ended, rankable games. These games, including both board games and card games, provide LLM agents with a peer-learning environment and unlimited upper bound for improvement. They enable agents to continually improve and compete, ensuring that the evaluation framework remains challenging as agent capabilities grow. Furthermore, our competitive arena is inherently extensible and can be readily adapted to other types of open-ended, rankable tasks, facilitating the assessment of core agent abilities in new domains. As agent capabilities continue to advance, CATArena can evolve by incorporating tasks with greater complexity and discrimination, thereby supporting ongoing evaluation without the need for expert-level human annotation.

In our experiments, we conduct comparative performance evaluations and data analysis conducted on our self-developed minimal code agent and state-of-the-art commercial code agents. CATArena consistently provides stable and reliable benchmarks for assessing both agent capabilities and the agentic potential of the underlying LLMs. Within the peer-learning framework, we design general scoring metrics to systematically assess the fundamental abilities of participating agents, including their in-context learning ability. Our experiments demonstrate that the strategy coding tasks applied in CATArena are fundamentally different from traditional LLM reasoning tasks. This represents a novel evaluation dimension that has not been addressed in previous work. Additionally, we analyze characteristics of CATArena, demonstrating its reliability and extensibility as a benchmarking platform.

In summary, our contributions are as follows:

- **Iterative Peer-learning-based Competitive Framework**: We propose a novel framework that leverages iterative peer-learning and competition to evaluate the learning abilities of LLM agents. Agents continuously revise their strategies based on feedback and outcomes from previous rounds, aligning agent evolution with human evolution.

- **CATArena Benchmark**: We introduce CATArena, a tournament-style benchmark for evaluating the basic capabilities of LLM agents using a diverse set of open-ended games, including board and card games. CATArena provides an unlimited upper bound for agent improvement and supports extensible evaluation across diverse, open-ended tasks.

- **Comprehensive Agent Evaluation**: We design general and systematic evaluation matrices and conduct comparative experiments and analyses between our minimal code agent and state-of-the-art commercial agents, demonstrating the reliability, stability, and extensibility of the CATArena.

## 2 RELATED WORK

**Learning Ability.** Learning ability is crucial for LLM agents, as it enables continual adaptation and improvement in dynamic and complex environments. In the context of our work, learning ability refers to the capacity of agents to optimize strategies using historical context, environmental feedback, and peer interactions without updating model parameters. Recent studies have shown that self-learning methods, such as self-refinement (Madaan et al., 2023; Shinn et al., 2023), allow models to enhance their outputs through iterative feedback, while environmental feedback further supports continual learning (You et al., 2024). In addition to self-learning, peer-learning has also been increasingly recognized, with approaches encouraging agents to learn from others' reasoning processes and shared experiences (Liang et al., 2024; Luo et al., 2025). These diverse learning mechanisms have led to notable advances in tasks such as code generation, complex reasoning, and collaborative problem-solving. In the context of LLM-driven agents, learning ability represents a critical capability that supports effective adaptation and enables agents to tackle increasingly complex tasks and evolving challenges (Zhu et al., 2025; Gao et al., 2025). Despite the progress,

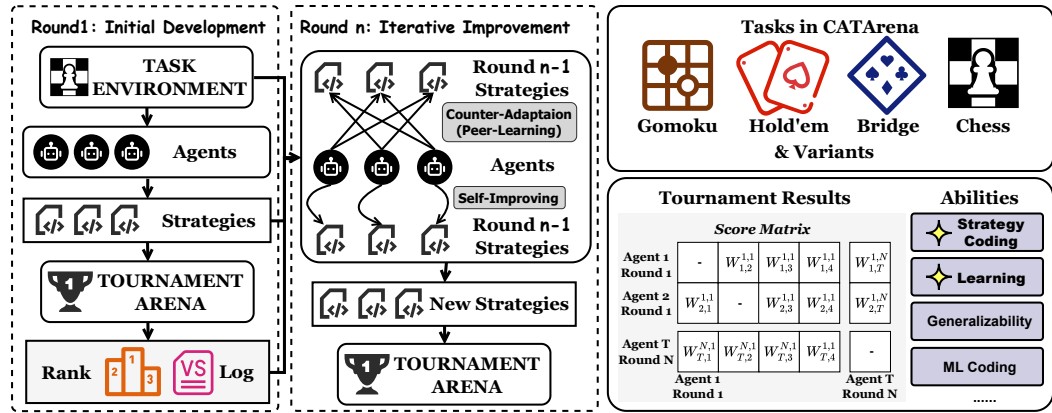

Figure 1: **Overview of the evaluation framework and CATArena**. The evaluation framework adopts an iterative peer-learning based competitive process. In round 1, LLM agents develop initial strategies via coding. These strategies are matched in a tournament arena, producing rankings and logs. In each subsequent round, agents analyze previous codes and logs, refine their strategies, and compete again. CATArena includes four open-ended and rankable games to cover diverse settings. Based on the tournament results from all rounds, a carefully designed scoring matrix and evaluation scheme are used to robustly quantify various agent abilities.

systematic evaluation of how agents learn from each other remains underexplored, highlighting the need for benchmarks that capture both self-learning and peer-learning abilities.

**Evaluation on Agents.** Recent benchmarks primarily assess LLM-driven agents on end-to-end, task-specific abilities. Code-based evaluations such as GitTaskBench (Ni et al., 2025), SUPER (Bogin et al., 2024), ProjectEval (Liu et al., 2025), SWE-PolyBench (Rashid et al., 2025), Red-Code (Guo et al., 2024), SWT-Bench (Mündler et al., 2025), InfiAgent-DABench (Hu et al., 2024), and DA-Code (Huang et al., 2024) focus on large-scale software development, code security, bug fixing, and data science tasks. Other works extend agent evaluation to research (Du et al., 2025), real-world tool use (Yao et al., 2024), and assistant scenarios (Mialon et al., 2024). While some benchmarks explore agent-vs-agent evaluation (Zhuge et al., 2024), most rely heavily on human annotation and objective correctness, leading to upper bounds and saturation as agent capabilities advance.

**Open-ended Tasks.** To address these limitations, recent work has leveraged open-ended, rankable tasks. Benchmarks such as GameBench (Costarelli et al., 2024), lmgame-Bench (Hu et al., 2025a), GAMEBot (Lin et al., 2024), and card game evaluations (Wang et al., 2025) assess LLMs' strategic reasoning through diverse games. Frameworks like Game Reasoning Arena (Cipolina-Kun et al., 2025), GVGAI-LLM (Li et al., 2025), ZeroSumEval (Alyahya et al., 2025), TextArena (Guertler et al., 2025), and MCU (Zheng et al., 2025) further extend evaluation to multi-turn reasoning, spatial adaptability, natural language interaction, and open-ended tasks. However, these benchmarks mainly focus on reasoning skills and do not systematically evaluate agents' learning abilities or coding strategies. Our analysis shows that measuring learning ability and coding strategy is fundamentally distinct, and both are essential for advancing agent intelligence. Additionally, in human board game competitions, variant rules such as Chess960 (FIDE, 2023) and Six-plus Hold'em (sixplusholdem, 2025) are often introduced to reduce memorization and encourage creativity. Notably, these variant rules have received relatively less attention in model evaluation.

Table 1: Overview of game arenas and representative variants in CATArena.

| Game | Symmetry | Type | Players | Variant |
|------|----------|------|---------|---------|
| Gomoku | ✓ | Board | 2 | Forbidden points; dual three-in-a-row |
| Texas Hold'em | ✗ | Card | $\geq 8$ | Card removal; swapped hand ranks |
| Chess | ✓ | Board | 2 | Chess960; forbidden/special moves |
| Bridge | ✓* | Card | 4 | Card exchange |

*  For Bridge, symmetry is defined by assigning identical agent strategies to both teammates.

## 3 CATARENA

### 3.1 ITERATIVE PEER-LEARNING BASED COMPETITIVE FRAMEWORK

As shown in Figure 1, we propose an iterative peer-learning-based competitive framework, where CATArena evaluates code agents through a two-phase workflow: *initial strategy development* and *iterative improvement*. The initial phase assesses each agent's ability to independently implement a baseline strategy based on the game code, while the iterative phase focuses on the agent's learning ability.

**Initial Development (Round 1).** In this stage, each agent receives the game code and a sample AI implementation. Without external guidance, each agent must develop its own strategy to participate in the tournament. This phase primarily examines the agent's strategy coding ability and establishes a baseline for subsequent evaluation.

**Iterative Improvement (Rounds $n > 1$).** After the first round, all strategies submitted are evaluated through a tournament (round-robin format for symmetric games, batch-based competition for asymmetric games). Comprehensive competition logs are generated, recording rankings, win counts, and move histories for all matches. In subsequent rounds, agents are provided exclusively with the game code, all submissions, and detailed logs from the immediately preceding round only. This design strictly restricts agents to develop their new strategies based solely on the information from the previous round, thereby preventing access to the entire historical record and effectively controlling the context length. By limiting the amount of input information, this approach avoids excessive context that could negatively impact model performance. Agents must analyze these resources from previous rounds to adapt and improve their own strategies. This phase assesses the agent's learning ability through repeated cycles of analysis and refinement.

This iterative evaluation framework of CATArena enables a granular assessment of both basic coding skills and advanced learning capabilities, supporting a robust and scalable measurement of code agent performance.

### 3.2 GAMES AND VARIANTS

Building on the tournament-based evaluation framework, CATArena deploys four distinct game arenas, each selected to test the strategic reasoning and coding capabilities of code agents across varying levels of complexity and interaction patterns. These arenas include competitive and cooperative settings, as well as symmetric and asymmetric game structures, thus enabling a diverse analysis of agents' strategy coding abilities and learning patterns.

In addition to standard rules, each game is extended with thoughtfully designed variants that introduce novel or altered mechanics, inspired by real-world adaptations such as Fischer Random Chess (Chess960) (FIDE, 2023). Like human competition, the variant rules encourage strategy generalization and penalize rote memorization, as most models are trained on card and board game data. Table 1 provides an overview of the selected games and their respective variants.

### 3.3 TOURNAMENT FORMAT AND SCORING SYSTEM

After the completion of all $N$ development rounds, CATArena conducts a comprehensive tournament to quantitatively evaluate agent strategies and compute performance metrics. A total of $T$ agent models participate, each contributing strategies in every round of development. Tournament formats are tailored to game types: for symmetric games, all strategies engage in a round-robin

cycle, ensuring exhaustive pairwise competition; for asymmetric games such as Texas Hold'em, strategies are grouped into batches and compete in multi-agent matches. To mitigate randomness, all matches are repeated multiple times, and results are averaged for robust evaluation.

Scores are recorded in a scoring matrix $W \in \mathbb{R}^{(TN) \times (TN)}$, where $W_{i,j}^{n,m} \in [0, 1]$ denotes the score obtained by agent $i$'s strategy in round $n$ against agent $j$'s strategy in round $m$. When $n = m$, the notation simplifies to $W_{i,j}^n$; similarly, when $i = j$, it is denoted as $W_i^{m,n}$. This scoring system enables fine-grained, quantitative analysis of agent performance in both individual and iterative development stages. For asymmetric games, pairwise results are not feasible; instead, batch-based tournaments are used and the score matrix records the win rates of multi-agent matches. The tournament format is provided in the Appendix A.

## 3.4 EVALUATION METRICS

Based on the scoring matrix $W$, we design a set of evaluation metrics to quantitatively assess the key capabilities of code agents. Specifically, our metrics are constructed to measure three core capabilities: *strategy coding*, *learning*, and *generalizability*. In the following sections, we define these metrics using symmetric games as examples. For asymmetric games, the evaluation principles remain consistent. The calculation of the scoring matrix $W$ is provided in Appendix B.

**Strategy Coding.** Strategy coding measures the agent's fundamental ability to abstract game strategies into reproducible algorithms and implement them as executable code, which is fundamentally different from general reasoning and strategic planning abilities. In CATArena, this metric evaluates how effectively an agent can independently develop a baseline strategy for the game environment and compete against other agents in the initial development stage.

For each agent $i$, strategy coding is quantified by the average score obtained against all other agents in the first round:

$$S_i = \operatorname{avg}_{j \neq i}(W_{i,j}^1).$$

This metric serves as the foundational benchmark for code agent evaluation in CATArena.

**Learning Ability.** The learning capability of a code agent captures its ability to leverage historical information and opponent behaviors to improve its own performance.

*Global Learning* assesses an agent's overall improvement in strategy quality. This metric evaluates the relative performance of agent $i$'s strategies against all strategies from all agents and rounds, and measures the average progress made compared to its initial baseline. It serves as the primary indicator of learning ability.

Formally, for agent $i$, global learning is defined as:

$$L_i = \operatorname{average}_{n=2}^N \left( G_i^n - G_i^1 \right),$$

where $G_i^n$ represents the global performance of agent $i$'s strategy from round $n$:

$$G_i^n = \operatorname{average}_{(i,n) \neq (j,m)} \left( W_{i,j}^{n,m} \right).$$

This metric captures the agent's ability to learn and adapt over multiple rounds, reflecting its progress in a comprehensive competitive landscape.

*Counter-Adaptation* measures an agent's targeted learning ability, reflecting its capacity to achieve improved results against opponents in successive rounds. For agent $i$, the counter-adaptation score is defined as the average improvement in scores against other agents from round $n - 1$ to round $n$ ($n \geq 2$):

$$C_i = \operatorname{average}_{n=2}^N \left( A_i^n - B_i^{n-1} \right),$$

where the advance score $A_i^n$ and base score $B_i^{n-1}$ are defined as:

$$A_i^n = \operatorname{average}_{j \neq i} \left( W_{i,j}^{n,n-1} \right), B_i^{n-1} = \operatorname{average}_{j \neq i} \left( W_{i,j}^{n-1} \right).$$

Here, $A_i^n$ represents agent $i$'s average performance in round $n$ against the strategies submitted by other agents in the previous in round ($n - 1$). The base score $B_i^{n-1}$ denotes agent $i$'s average

performance in round $n - 1$ against those same opponents. This comparison isolates the agent's targeted adaptation from one round to the next.

*Self-improvement* evaluates an agent's capacity to genuinely enhance its strategies over successive rounds of development. This metric reflects whether newly developed strategies can consistently outperform the agent's own previous versions.

We quantify self-improvement by calculating the Pearson correlation Pearson (1896) between the round index and the agent's average scores across rounds. For agent $i$, the self-improvement score is defined as

$$\text{SI}_i = \text{Pearson}\left([1, \cdots, N], \ [S_i^1, \cdots, S_i^N]\right).$$

Here, $S_i^n$ denotes the average score of agent $i$'s strategy from round $n$ against its own strategies from other rounds

$$S_i^n = \text{average}_{m \neq n}\left(W_i^{n,m}\right).$$

A higher self-improvement score indicates a stronger ability to iteratively refine and upgrade strategies throughout the development process.

**Generalizability.** Generalizability measures an agent's ability to comprehend and adapt to novel or altered game rules that differ from those encountered during training or prior experience. This metric specifically evaluates the agent's capacity to generalize beyond previously seen environments, focusing on handling new or modified scenarios. For agent $i$, the generalizability score is defined as:

$$U_i = B_i^{1;\text{Variants}} - B_i^{1;\text{Standard}},$$

where $B_i^{1;\text{Variants}}$ and $B_i^{1;\text{Standard}}$ denote the base scores of agent $i$ in the first round under variant and standard rule settings, respectively. A higher value of $U_i$ indicates stronger generalizability, reflecting the agent's ability to effectively develop and apply strategies for previously unseen tasks.

## 4 EXPERIMENTS

### 4.1 EXPERIMENTAL SETUPS

**Participants.** In our experiment, we employ three types of agents: (1) **Minimal Agents (LLM + ADK Framework):** A baseline agent developed with the Agent Development Kit (ADK) Python toolkit. We provide essential tools, including file manipulation, bash scripting, and Python execution, for the ADK code agent. On this foundation, we integrate state-of-the-art LLMs to systematically compare their core competencies as code agents for strategy implementation. (2) **Commercial Code Agents:** State-of-the-art and commercial CLI-based agents (e.g., Claude Code, CodeX, Gemini-CLI, Qwen-Coder) are included for benchmarking. These agents feature advanced integration with various command-line interfaces, tools, and LLMs, resulting in enhanced overall capabilities. These agents serve as leading solutions in code agent development and provide valuable reference points for future research. (3) **LLM-Player:** In this control setting, LLMs directly output game moves without generating code. For each turn, the LLM receives the game rules, current state, and history, and returns the next action. This approach is specifically designed to assess the inherent strategic and reasoning capabilities of LLMs. Detailed agent parameter settings and model selections are presented in Appendix C.

**Tournaments.** All experiments are conducted under two main tournament settings: (1) a comparison among minimal agents equipped with different LLMs ($T_1 = 6$), and (2) a comparison between the best-performing minimal agent and a set of commercial code agents ($T_2 = 5$). To reduce the impact of randomness on strategy generation, each tournament is repeated for four times, and all reported metrics are averaged over the four runs. Each tournament consists of $N = 4$ rounds of iterative development. To further mitigate stochastic effects in competition outcomes, every entry in the scoring matrix $W$ is estimated by repeated matches. Detailed tournament prompts are listed in Appendix M.

We report the detailed scoring policy, generation configs, and repetition experiments in Appendix D. It is noteworthy that agents tend to generate different codes in repeated experiments, but their rankings are relatively stable.

Table 2: Agent Specifications and Open-Source Status.

| Agent Type | Agent Framework | Model | Agent OSS | LLM OSS |
|---|---|---|---|---|
| Minimal | basic code tools with ADK framework | DeepSeek-3.1 (DeepSeek-AI, 2024) | ✓ | ✓ |
| | | Qwen3-Coder-480B (Team, 2025c) | ✓ | ✓ |
| | | Doubao-Seed-1.6 (Team, 2025d) | ✓ | ✗ |
| | | GPT-5 (OpenAI, 2025b) | ✓ | ✗ |
| | | Claude-4-Sonnet (Anthropic, 2025a) | ✓ | ✗ |
| | | Gemini-2.5-pro (Team, 2025a) | ✓ | ✗ |
| Commercial | Gemini-CLI (Google, 2025) | Gemini-2.5-pro (Team, 2025a) | ✓ | ✗ |
| | Claude-Code (Anthropic, 2025b) | Claude-4/3.7 Hybrid (Anthropic, 2025a) | ✗ | ✗ |
| | CodeX (OpenAI, 2025a) | GPT-5 (OpenAI, 2025b) | ✓ | ✗ |
| | Qwen-Coder (Team, 2025b) | Qwen3-Coder-480B (Team, 2025c) | ✓ | ✓ |
| Other | LLM-Player | Agents's Corresponding LLM | N/A | N/A |

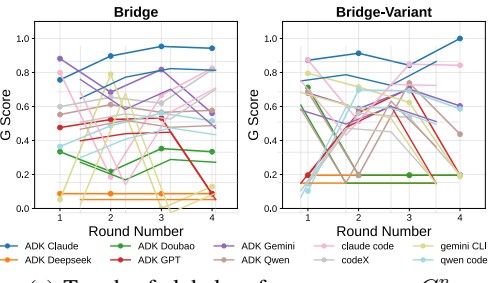

(a) Trends of global performance scores $G_i^n$.

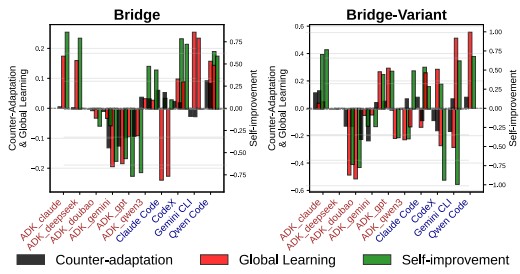

(b) Quantitative comparison of learning abilities.

Figure 2: **Visualization of agents' learning patterns and scores.**. For clarity, we use family names to represent LLM models instead of their full names. The results for other games are in Appendix E.

## 4.2 MAIN RESULTS

**Learning Ability.** Figure 2(a) visualizes the global performance scores $G_i^n$ revealing overall trends in agent strategies across iterations. Some agents, such as minimal agents driven by claude, exhibit a clear upward trajectory over multiple rounds, demonstrating strong learning capability. However, the performance of most agents remains unstable, and no obvious trend is observed. To explicitly illustrate the learning ability of each agent, we design a quantitative scoring method for global learning and introduce two additional learning modes: counter-adaption and self-improvement. Counter-adaptation measures an agent's targeted adaptation to specific opponent strategies, while self-improvement quantifies the optimization of its own consecutive strategies, reflecting self-evolution capability.

The quantitative results for these three abilities are presented in Figure 2(b). Compared to Figure 2(a), this decomposition offers finer-grained insights into whether an agent's progress is primarily driven by adapting to opponents or by refining its own strategies.. When both counter-adaption and self-improvement scores are positive, it indicates that the agent can effectively learn from both its opponents and itself, resulting in a positive global learning score. Compared to minimal agents, commercial agents exhibit stronger learning capabilities. Learning ability results for other games and a case study of strategy iteration mechanisms are provided in Appendix E. Through an analysis of the consistency of agent actions in endgame states, we observe that a majority of agents

Table 3: **Main LeaderBoard of CATArena.** We conduct two groups of tournaments between minimal agents (rank from 1 to 6) and commercial agents (rank from 1 to 5), and report the average ranking across all four tasks. Metrics include S.C. (Strategy Coding), G.L. (Global Learning), and G.A. (Generalizability).

| | Agents | Standard | | Variant | | G.A. ↓ |
|---|---|---|---|---|---|---|
| | | S.C. ↓ | G.L. ↓ | S.C. ↓ | G.L. ↓ | |
| Minimal | Claude-4-Sonnet | **1.25** | **2.5** | **1.75** | 2.75 | 5.00 |
| | DeepSeek-Chat | 5.75 | 2.75 | 4.25 | **2.75** | 2.75 |
| | Doubao-Seed | 3.75 | 4.75 | 3.75 | 4.50 | 2.75 |
| | Gemini-2.5-Pro | 3.25 | 3.75 | 3.25 | **2.75** | 3.25 |
| | GPT-5 | 3.75 | 3.50 | 3.00 | 3.75 | **2.25** |
| | Qwen3-Coder | 2.25 | 3.75 | 3.00 | 4.5 | 4.75 |
| Commercial | best ADK | 3.25 | **2.25** | **2.00** | 3.75 | 2.50 |
| | Claude-Code | 2.50 | 3.75 | 2.50 | 2.75 | 3.25 |
| | CodeX | **2.25** | 2.75 | 3.00 | 3.00 | 3.25 |
| | Gemini-CLI | 3.50 | **2.25** | 3.00 | 4.00 | **2.00** |
| | Qwen-Coder | 3.00 | 3.75 | 4.00 | **1.25** | 3.25 |

Table 4: **Main results of CATArena.** We conduct two groups of tournaments between Minimal agents and commercial agents. For each tournament, we display the results for Strategy Coding (S.C.↑), Global Learning (G.L.↑), and Generalizability (G.A.↑). The S.C.↑ score ranges from 0 to 1, a G.L.↑ score greater than 0 indicates the agent has learning ability, and the G.A.↑ score ranges from −1 to 1. All scores represent the relative performance of participants within the tournament.

| Games | | Gomoku | | | | | Hold'em | | | | | Bridge | | | | | Chess | | | | |
|---|---|---|---|---|---|---|---|---|---|---|---|---|---|---|---|---|---|---|---|---|---|
| | | Standard | | Variant | | | Standard | | Variant | | | Standard | | Variant | | | Standard | | Variant | | |
| Agents | | S.C.↑ | G.L.↑ | S.C.↑ | G.L.↑ | G.A.↑ | S.C.↑ | G.L.↑ | S.C.↑ | G.L.↑ | G.A.↑ | S.C.↑ | G.L.↑ | S.C.↑ | G.L.↑ | G.A.↑ | S.C.↑ | G.L.↑ | S.C.↑ | G.L.↑ | G.A.↑ |
| Minimal | Claude-4-Sonnet | **0.88** | -0.447 | **0.78** | -0.156 | -0.14 | **0.58** | 0.118 | 0.13 | **0.110** | -0.45 | 0.79 | **0.174** | **1.0** | 0.047 | 0.005 | **0.90** | -0.170 | 0.65 | 0.018 | -0.55 |
| | Deepseek-Chat | 0.23 | 0.027 | 0.38 | 0.077 | 0.13 | 0.01 | 0.010 | 0.00 | -0.022 | -0.01 | 0 | 0 | 0.10 | 0 | 0.10 | 0 | 0 | 0.10 | **0.049** | 0.10 |
| | Doubao-Seed | 0.33 | -0.192 | 0.72 | -0.302 | **0.46** | 0.04 | -0.035 | 0 | 0 | -0.04 | 0.2 | -0.033 | 0.45 | -0.516 | 0.40 | 0.58 | -0.337 | 0.10 | 0.034 | -0.46 |
| | Gemini-2.5-Pro | 0.25 | -0.066 | 0.00 | **0.173** | -0.12 | 0.01 | 0.020 | 0.00 | 0.078 | -0.01 | **0.90** | -0.195 | 0.60 | -0.049 | -0.30 | 0.58 | -0.147 | **0.90** | 0.003 | **0.46** |
| | GPT-5 | 0.48 | **0.062** | 0.76 | -0.019 | 0.18 | 0.16 | **0.102** | 0.87 | -0.050 | **0.71** | 0.47 | -0.095 | 0.10 | **0.293** | -0.02 | 0.38 | -0.525 | 0.45 | -0 | 0.24 |
| | Qwen3-Coder | 0.85 | -0.523 | 0.36 | -0.089 | -0.50 | 0.20 | 0.038 | 0.00 | 0.003 | -0.20 | 0.65 | 0.032 | 0.76 | -0.230 | -0.19 | 0.58 | -0.187 | 0.80 | -0.532 | 0.21 |
| Commercial | best ADK | 0 | 0.075 | **0.75** | -0.022 | **0.88** | 0.07 | **0.073** | 0.46 | 0.030 | **0.39** | 0.25 | **0.295** | 0 | 0.361 | -0.25 | **0.91** | -0.110 | **1.00** | -0.342 | -0.47 |
| | Claude-Code | 0.78 | -0.322 | 0.66 | **0.194** | -0.28 | 0.01 | 0.100 | 0 | 0.105 | -0.001 | **1.00** | -0.240 | **0.93** | -0.139 | -0.04 | 0.56 | -0.158 | 0.44 | -0.226 | 0.03 |
| | CodeX | 0.47 | **0.454** | 0.69 | -0.095 | 0.34 | **0.72** | 0.050 | 0.17 | 0.067 | -0.55 | 0.75 | 0.098 | 0.50 | 0.285 | -0.25 | 0.38 | 0.033 | 0.34 | **0.064** | 0.09 |
| | Gemini-CLI | 0.31 | 0.260 | 0.19 | 0.172 | 0.0 | 0.13 | 0.050 | 0.37 | -0.007 | 0.24 | 0.01 | 0.254 | 0.83 | -0.286 | **0.79** | 0.38 | **0.395** | 0.38 | -0.154 | 0.16 |
| | Qwen-Coder | **0.94** | -0.054 | 0.22 | -0.530 | -0.94 | 0.07 | 0.058 | 0 | **0.105** | -0.007 | 0.49 | 0.157 | 0.25 | **0.556** | -0.25 | 0.28 | -0.204 | 0.34 | 0.039 | **0.19** |

indeed learn from the code generated in the previous round, leading to increasing consistency. This phenomenon is mos

**Ability Evaluation.** We conduct two sets of tournaments: (1) minimal agents equipped with different LLMs, (2) the best-performing minimal agent against commercial code agents. For each tournament, we report the average ranking and scores for three core agent capabilities. The main leaderboard and main results of CATArena are summarized in Table 3 and Table 4. These results reveal the following key observations:

**Observation 1: The performance gap among LLMs is more pronounced in minimal agents compared to commercial agents.** Table 3 shows that Claude-4-Sonnet achieves the highest score among minimal agents, while the rankings of other LLMs are more dispersed. In contrast, commercial agents driven by the same LLMs exhibit much closer average rankings, with all agents scoring around 2.5 out of 5, indicating a reduced performance gap. Moreover, commercial agents demonstrate performance levels similar to the best-performing minimal agent. This suggests that *the underlying agent framework can significantly influence how effectively an LLM's capabilities are utilized*, as commercial agents are often optimized for specific models.

**Observation 2: The participating agents display different ranking orders across various capabilities.** The tournament results reveal that the relative rankings of agents change depending on the specific core ability being tested. The ranking of these abilities provides a decomposition of end-to-end performance, offering insights for further optimization of both LLMs and agent frameworks.

**Observation 3: Agents exhibit varied performance distributions across different tasks.** The results indicate that agents' performance is not uniformly distributed across all tasks, which is mainly attributed to the distinct nature and difficulty of the four tasks. In the variant tasks, the performance gap among agents is more pronounced, likely because the game rules and strategies are less familiar to agents.

## 4.3 EFFECTIVENESS OF CATARENA

We design a series of experiments to demonstrate the effectiveness of CATArena.

**Comparison between Agents and LLM-Players.** The primary task in CATArena is strategy coding, which relies on the underlying coding capabilities of LLMs. We posit that reasoning over code to develop strategies is fundamentally different from direct reasoning during gameplay. To validate this distinction, we compare agent-developed strategies with the LLM-Player baseline (see Appendix F). Our results show that current agents primarily implement simple rule-based algorithms, indicating substantial room for advancement in agents' strategy coding abilities. CATArena fully leverages this non-saturation, enabling sustainable iterative peer-learning.

Table 5: **Collective learning trends of agents across different tasks in CATArena.** $DIS_{range}$ and $DIS_{std}$ represent the Pearson correlation coefficients of the range and standard deviation of agent performance scores over four rounds, reflecting the similarity and dispersion of agent strategies. $Trend_{mean}$ denotes the Pearson correlation between the mean agent performance and the round number, indicating the overall trend of group improvement.

|  | Gomoku | | Hold'em | | Bridge | | Chess | |
|---|---|---|---|---|---|---|---|---|
|  | StdV. | VarV. | StdV. | VarV. | StdV. | VarV. | StdV. | VarV. |
| $DIS_{std}$ | -0.05 | 0.15 | -0.81 | -0.80 | -0.82 | -0.57 | 0.55 | -0.04 |
| $DIS_{range}$ | -0.16 | 0.44 | -0.80 | -0.76 | -0.54 | -0.33 | -0.08 | 0.16 |
| $Trend_{mean}$ | 0.42 | -0.02 | 0.75 | 0.67 | 0.24 | -0.10 | -0.74 | -0.79 |

To further analyze the similarities and differences between agent-implemented code strategies and those of LLM-Players, we ask agents' code and LLM-Player to select the next action on endgame states. Figure 4.3 illustrates the action consistency between agents' code and LLM-Players in Chess. Surprisingly, *the strategies encoded in agent code differ significantly from those inferred directly by the LLM*, even if they are from the same model. Meanwhile, strategies produced by different agents and different LLMs also show notable similarities. This indicates that strategy coding and reasoning in LLMs are distinct capabilities. We report results of other tasks, as well as the case study of strategy comparison in Appendix F and G. CATArena evaluates the strategy coding ability of agents rather than their reasoning ability, thereby filling a gap in previous benchmarks. The relationship between strategy coding ability and LLM-based strategy reasoning remains unclear and requires further investigation.

**Collective Learning Trends Among Agents.** We analyze the collective learning dynamics of agents across tasks, as presented in Table 5. The metrics $DIS_{range}$ and $DIS_{std}$ report the Pearson correlation between the standard deviation and range of agent performance scores over four rounds ($B_i^n, n = 1, 2, 3, 4$). The higher similarity in performance scores (i.e., DIS > 0) suggests that agents can learn effective strategies more readily, indicating lower task difficulty. Based on these results, the relative difficulty ranking of tasks in CATArena is Chess > Gomoku > Bridge > Hold'em.

$Trend_{mean}$ captures the trend in the average performance of all agents, $\text{average}_i(G_i^n)$, across rounds (calculated as the Pearson correlation between the mean score and the number of rounds). Our analysis reveals that agents are able to col-

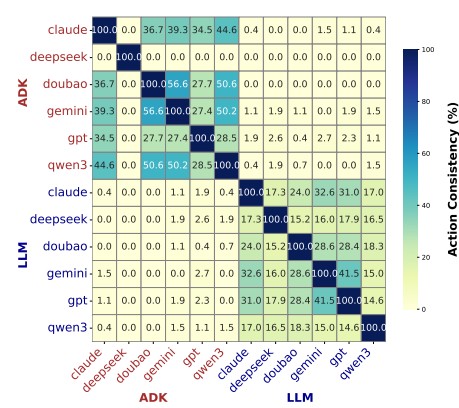

Figure 3: Action consistency between agents' code and LLM-Players on Chess endgames.

lectively improve their strategies in simpler environments, whereas their learning capacity remains limited in more challenging tasks. Furthermore, the collective improvement observed in variant tasks is lower than in standard versions, indicating that variants present greater difficulty for current agents.

**Agent Evolution Over Extended Rounds** To investigate the long-term learning behavior of the models, we conduct additional rounds of experiments to explore the agents' extended learning trends in Appendix H. The results show that the first four rounds are the most active phases for agent learning and strategy optimization. Afterward, rankings of models gradually stabilize, indicating the emergence of a performance plateau.

**Additional Results.** CATArena's iterative peer-learning framework is easily extensible to new tasks for evaluating other fundamental agent abilities. We demonstrate this by introducing a Machine Learning (ML) track and multi-lingual track, with experimental results provided in Appendix I and Appendix J, respectively. Experimental results indicate that current agents still exhibit substantial potential for improvement. As agents continue to advance, the open-ended task design and peer-

learning evaluation framework of CATArena ensure that systematic assessment can be sustained over time. We also report agent cost in terms of token usage, time consumption, and generated code statistics for each agent, in Appendix L. Notably, Claude-4-Sonnet utilizes the most tools and tokens, and also develops a significantly larger amount of code. In contrast, GPT-5 achieves the best balance between token usage and performance. Efficient utilization of tokens and external tools remains an important research direction to advance the capabilities of LLM agents. We also report HTTP error rates of generated code in Appendix K, providing an additional perspective for evaluating LLM agents. The results show that commercial agents outperform minimal agents in terms of code implementation completeness. Furthermore, implementing standard games is generally easier for LLM agents compared to variant games.

## 5 CONCLUSION

In this work, we address two fundamental challenges in LLM agent evaluation: the need for systematic measurement of learning ability, and the tendency of traditional benchmarks to become saturated as agent capabilities improve. To this end, we propose an iterative peer-learning-based competitive framework, enabling agents to continually revise and enhance their strategies through dynamic interaction and feedback. Building on this, we introduce CATArena, a tournament-style benchmark featuring open-ended and rankable board and card games. CATArena provides an environment with unlimited potential for agent improvement potential and extensible evaluation across new domains. Experimental results demonstrate that our framework reliably assesses core agent abilities, particularly learning ability and strategy coding, while ensuring stability and scalability. The open and flexible architecture of CATArena supports ongoing research and benchmarking for future intelligent agents.

**Limitations.** The current evaluation in CATArena is limited to four games, which primarily assess agents' learning ability and strategy coding. These scenarios do not encompass the full spectrum of potential LLM agent capabilities. In future work, we plan to introduce a wider variety of more complex tasks to evaluate agents' learning and other abilities from different perspectives.

## ETHICS STATEMENT

Our methods and algorithms do not involve adversarial attacks and do not threaten human safety. We conduct all experiments in controlled environments and do not encounter ethical or fairness issues.

## THE USE OF LARGE LANGUAGE MODELS

We use large language models as general-purpose assistants for text editing, including grammar correction, wording and tone adjustment, punctuation, and stylistic consistency. The models do not contribute to research ideation, methodology, experimental design, data analysis, interpretation of results, or the creation of substantive academic content or references. We carefully review and approve all suggestions from the models, and we take full responsibility for the final manuscript.
We also use large language models to help write code for game development and visualization, with careful examine from the authors.
We do not use language models for data synthesis or augmentation.

## REPRODUCIBILITY STATEMENT

We provide the source code for this paper at `https://anonymous.4open.science/r/codeagentgame-2CB3` (Minimal agent) and `https://anonymous.4open.science/r/adk_catarena-DFB7` (arena).
We report extensive results from repeated experiments in the Appendix D.

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

## A  TOURNAMENT FORMAT AND SCORING SYSTEM

Table 6: Configs of tournament on four games.

| Environment | Code Agents | LLM-player |
|---|---|---|
| Gomoku | Board size: 15×15
Number of pairwise matches: 4 × 2
Swap black and white pieces after each match
Maximum time per move: 10 s | Board size: 15×15
Number of pairwise matches: 2 × 2
Swap black and white pieces after each match
Maximum time per move: 600 s |
| Texas Hold'em | Max players: 12
Rounds: 100
Random shuffle seat after each round
Initial chips: 2000
Blind increase every 24 hands
Max hands per round: 720 or until winner decided
Maximum time per move: 3 s | Max players: 12
Rounds: 10
Random shuffle seat after each round
Initial chips: 2000
Blind increase every 24 hands
Max hands per round: 720 or until winner decided
Maximum time per move: 1000 s |
| Bridge | Number of pairwise matches: 12 × 2
Swap directions of open/closed rooms
Use same deck for each pair of match
Maximum time per move: 10 s | Number of pairwise matches: 12 × 2
Swap directions of open/closed rooms
Use same deck for each pair of match
Maximum time per move: 200 s |
| Chess | Number of pairwise matches: 8 × 2
Swap black and white pieces after each match
Maximum moves per game: 200
Maximum time per move: 10 s | Number of pairwise matches: 2 × 2
Swap black and white pieces after each match
Maximum moves per game: 200
Maximum time per move: 600 s |

We list the basic settings for each games in Table 6.

For each game, we ensure that the number of pairwise matches among code agents allows the final results to stabilize, i.e., for each game, the L1-norm fluctuation of the scoring matrix $W$ is less than 5%.

Specifically, due to the inherent randomness in card dealing in Texas Hold'em, we conducted an additional analysis on the stability of the win rates. Figure 4 illustrates the cumulative win rate trends over 10 to 300 rounds for a representative tournament. To further assess the reliability of our evaluation, we analyze the win rate trends from different starting rounds to the final round. Based on this analysis, starting from round 50, the cumulative win rates of all models become stable, with the absolute value of the linear regression slope dropping below 0.0002 (i.e., per-round changes less than 0.02%). This indicates that our tournament design of 100 rounds, with each round consisting of 200 to 720 hands, is sufficient to cover the stochasticity of the game and provides a robust and reliable assessment of model performance. The results of the tournament reliably reflect the true capabilities of the evaluated models, ensuring the soundness of subsequent algorithm comparisons and optimizations.

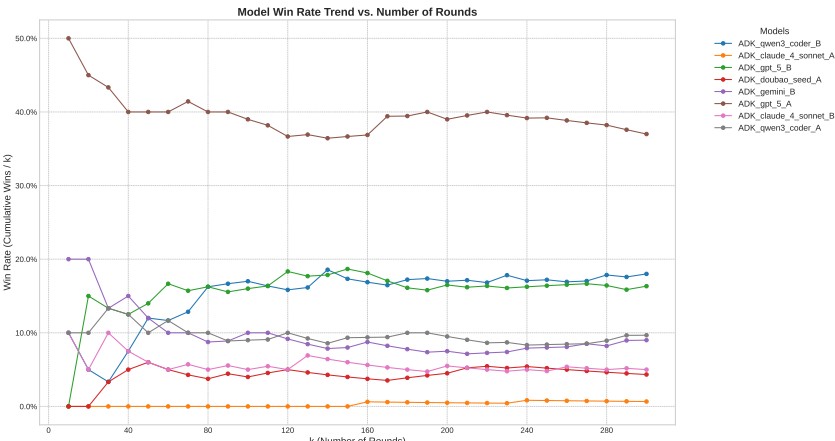

Figure 4: Cumulative win rate trajectories of Texas Hold'em agents over 300 tournament rounds.

For LLM-players, since their reasoning time is relatively long for most games, we reduce the number of repeated experiments. Note that our comparison with LLM-players is only a qualitative analysis

of the differences between LLM-player and corresponding coding agent. The exploration of LLM-players' results is not the focus of this paper; refer to the main text for details.

## B  EVALUATION METRIC CALCULATION

Table 7: Scoring rules of tournament on four games.

| Environment | Scoring Metric |
|---|---|
| Gomoku | Pairwise match scoring:
Win = 1 point,  Draw = 0.5 point,  Lose = 0 point. |
| Texas Hold'em | Multi-agent batches:
score is the average win rate across all tournaments participated. |
| Bridge | 20 VP system:
Two opposing pairs' scores sum to 20,
Final score divided by 20, ensuring each pair's score $\in [0, 1]$. |
| Chess | Pairwise match scoring:
Win = 1 point,  Draw = 0.5 point,  Lose = 0 point. |

To evaluate the basic capabilities of LLM agents, we define metrics for each applied game. Let $N$ be the number of rounds and $K$ be the number of participating agents. We construct a matchup matrix

$$W \in \mathbb{R}^{(N \cdot K) \times (N \cdot K)},$$

where the generic element $W_{i,j}^{n,m}$ denotes the score when agent $i$ from round $n$ plays against agent $j$ from round $m$. We abbreviate $W_{i,j}^{n}$ for same-round comparisons ($n = m$) and $W_{i}^{n,m}$ for self-comparisons across rounds ($i = j$). Diagonal entries $(n, i) = (m, j)$ are ignored.

For asymmetric games, pairwise results are not feasible; instead, batch-based tournaments are used and the score matrix records the win rates of multi-agent matches. For each batch, we obtain a single score group $W_{i_1,i_2,...,i_{BS}}^{n_1,n_2,...,n_{BS}}$, where $BS$ is the batch size. We conduct three types of experiments to accommodate different metric calculations: (1) $W_i^{1,2,...,N}$, where the same agent's strategies from $N$ rounds compete against each other, used to compute self-improvement metrics $S_i^n$; (2) $W_{i_1,i_2,...,i_T}^{n}$, where all agents in the same round compete, used to calculate the base score $B$; (3) All $N \times T$ agent strategies are randomly shuffled and grouped for competition (with $BS = 12$ in our experiments), used to compute both the global score $G_i^n$ and advanced score $A_i^n$.

Scoring rules for the four games are summarized in Table 7.

## C  GENERATION CONFIGS

For all LLMs used in our work, we set temperature to be $0.1$, max token identical to their official APIs' setting. We set top-p to $1.0$, Top-k to be $100$, and presence penalty to be default to the API.

Additionally, both Claude-4-Sonnet and DeepSeek-3.1 occasionally encounter tool call issues that result in no code being generated, as frequently reported by the community. If such errors occur three times in a row, we substitute Claude-4-Sonnet with Claude-3.7-Sonnet and DeepSeek-3.1 with DeepSeek v3.

For LLM-players, considering the uncertainty in model output formats, we allow up to three retries. The prompt of LLM-players are in arena's code and not present in paper considering its excessive length.

## D  REPETITION EXPERIMENTS

We report the results of repetition experiments($N = 4$) on first two tournament round in Table 8.

From table, we observe that 1. The rankings of most agents remain relatively stable ,with ranking standard deviation changes of less than one. However, a few agents, such as Gemini-2.5-Pro and Claude-Code, exhibit greater fluctuations; 2. The rankings for standard games are more stable than those for variant games; 3. The results of the open source model are more stable than those of the closed source model, and commercial agents are more stable than minimal agents; 4. Additionally,

Table 8: Standard deviation of ranking in Round 1 and Round 2 with repeating 4 times.

| Games | | Average | | Gomoku | | Hold'em | | Bridge | | Chess | |
|---|---|---|---|---|---|---|---|---|---|---|---|
| | | Standard | Variant | Standard | Variant | Standard | Variant | Standard | Variant | Standard | Variant |
| Round 1 — Minimal | Claude-4-Sonnet | 0.80 | 0.91 | 1.58 | 0.43 | 1.12 | 0.71 | 0.50 | 1.64 | 0.00 | 0.87 |
| | Deepseek-Chat | 0.72 | 0.81 | 0.83 | 0.87 | 0.83 | 1.22 | 1.22 | 0.71 | 0.00 | 0.43 |
| | Doubao-Seed | 1.58 | 0.90 | 1.87 | 1.87 | 1.09 | 0.43 | 2.06 | 0.87 | 1.30 | 0.43 |
| | Gemini-2.5-Pro | 1.24 | 1.23 | 1.50 | 1.30 | 1.12 | 0.83 | 1.12 | 1.66 | 1.22 | 1.12 |
| | GPT-5 | 0.75 | 1.18 | 0.71 | 1.50 | 0.71 | 1.30 | 1.09 | 1.50 | 0.50 | 0.43 |
| | Qwen3-Coder | 1.16 | 0.84 | 1.48 | 1.50 | 1.22 | 0.71 | 1.09 | 0.71 | 0.83 | 0.43 |
| Round 1 — Commercial | Claude-Code | 1.27 | 1.01 | 1.12 | 1.66 | 0.43 | 0.00 | 1.79 | 1.09 | 1.73 | 1.30 |
| | CodeX | 0.76 | 0.57 | 0.83 | 0.50 | 0.43 | 0.50 | 1.09 | 1.30 | 0.71 | 0.00 |
| | Gemini-CLI | 1.14 | 0.93 | 0.83 | 0.43 | 1.12 | 1.00 | 1.30 | 1.79 | 1.30 | 0.50 |
| | Qwen-Coder | 1.01 | 0.95 | 1.22 | 1.48 | 1.12 | 0.00 | 0.87 | 0.83 | 0.83 | 1.50 |
| Round 2 — Minimal | Claude-4-Sonnet | 0.81 | 0.55 | 1.48 | 0.50 | 0.83 | 0.43 | 0.50 | 0.83 | 0.43 | 0.43 |
| | Deepseek-Chat | 0.94 | 1.03 | 1.30 | 1.09 | 0.87 | 0.83 | 1.09 | 1.48 | 0.50 | 0.71 |
| | Doubao-Seed | 0.79 | 0.88 | 1.09 | 1.12 | 0.87 | 0.71 | 0.71 | 0.87 | 0.50 | 0.83 |
| | Gemini-2.5-Pro | 1.30 | 1.28 | 1.66 | 1.92 | 1.00 | 1.22 | 2.06 | 1.12 | 0.50 | 0.87 |
| | GPT-5 | 0.89 | 1.22 | 1.00 | 1.58 | 0.43 | 1.66 | 1.64 | 1.22 | 0.50 | 0.43 |
| | Qwen3-Coder | 0.82 | 1.02 | 1.58 | 1.30 | 0.83 | 0.43 | 0.43 | 1.50 | 0.43 | 0.83 |
| Round 2 — Commercial | Claude-Code | 1.20 | 1.04 | 1.09 | 1.41 | 1.22 | 0.43 | 1.64 | 1.09 | 0.83 | 1.22 |
| | CodeX | 0.71 | 0.87 | 0.43 | 0.87 | 0.50 | 1.09 | 0.83 | 1.09 | 1.09 | 0.43 |
| | Gemini-CLI | 0.88 | 1.10 | 0.83 | 1.64 | 0.83 | 0.83 | 1.41 | 1.48 | 0.43 | 0.43 |
| | Qwen-Coder | 1.13 | 0.90 | 1.66 | 0.83 | 0.43 | 0.83 | 1.00 | 1.12 | 1.41 | 0.83 |

we observe that agents do not consistently generate runnable code repositories across multiple development attempts. Even commercial agents occasionally fail to produce successful builds, which suggests that current code agents still need to improve their development stability.

# E LEARNING ABILITY

## E.1 GLOBAL LEARNING TREND

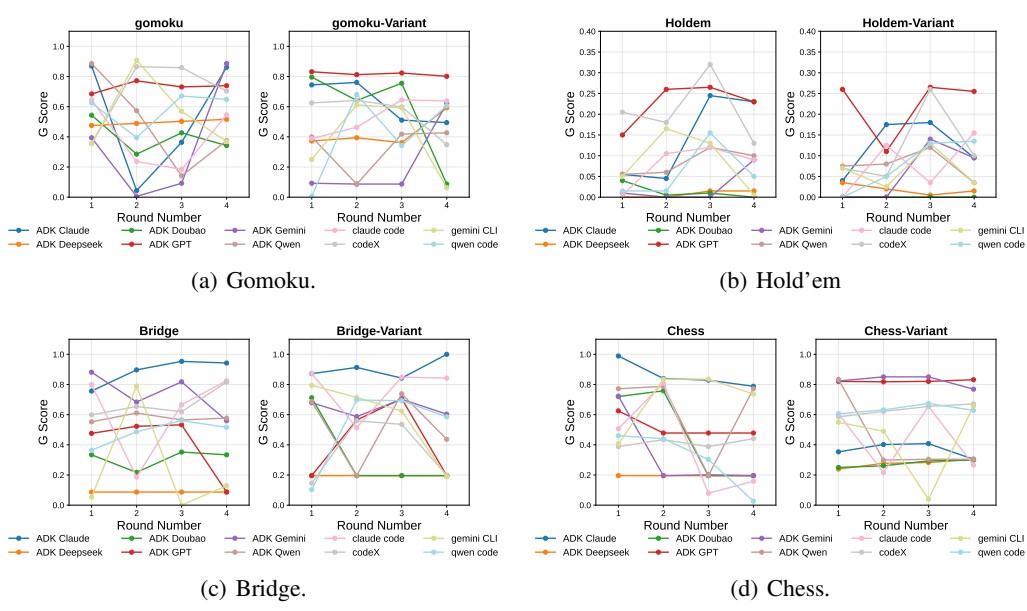

(a) Gomoku.

(b) Hold'em

(c) Bridge.

(d) Chess.

Figure 5: Trend of global performance score $G_i^n$ in Gomoku, Hold'em, Bridge and Chess.

Table 9: Global Learning with Group-wise Average Rankings.

| | Models | Avg. Ranking | | Gomoku | | Hold'em | | Bridge | | Chess | |
|---|---|---|---|---|---|---|---|---|---|---|---|
| | | Standard | Variant | Standard | Variant | Standard | Variant | Standard | Variant | Standard | Variant |
| Minimal | Claude-4-Sonnet | 2.50 | 2.75 | -0.447 | -0.156 | 0.118 | 0.110 | 0.174 | 0.047 | -0.170 | 0.018 |
| | Deepseek-Chat | 2.75 | 2.75 | 0.027 | 0.077 | 0.010 | -0.022 | 0.000 | 0.000 | 0.000 | 0.049 |
| | Doubao-Seed | 4.75 | 4.50 | -0.192 | -0.302 | -0.035 | 0.000 | -0.033 | -0.516 | -0.337 | 0.034 |
| | Gemini-2.5-Pro | 3.75 | 2.75 | -0.066 | 0.173 | 0.020 | 0.078 | -0.195 | -0.049 | -0.147 | 0.003 |
| | GPT-5 | 3.50 | 3.75 | 0.062 | -0.019 | 0.102 | -0.050 | -0.095 | 0.293 | -0.525 | -0.000 |
| | Qwen3-Coder | 3.75 | 4.50 | -0.523 | -0.089 | 0.038 | 0.003 | 0.032 | -0.230 | -0.187 | -0.532 |
| Commercial | best ADK | 2.25 | 3.75 | 0.075 | -0.022 | 0.073 | 0.030 | 0.295 | 0.361 | -0.110 | -0.342 |
| | Claude-Code | 3.75 | 2.75 | -0.322 | 0.194 | 0.100 | 0.105 | -0.240 | -0.139 | -0.158 | -0.226 |
| | CodeX | 2.75 | 3.00 | 0.454 | -0.095 | 0.050 | 0.067 | 0.098 | 0.285 | 0.033 | 0.064 |
| | Gemini-CLI | 2.25 | 4.00 | 0.260 | 0.172 | 0.050 | -0.007 | 0.254 | -0.286 | 0.395 | -0.154 |
| | Qwen-Coder | 3.75 | 1.25 | -0.054 | 0.536 | 0.058 | 0.105 | 0.157 | 0.556 | -0.204 | 0.039 |

Table 10: Counter-adaptation Score with Group-wise Average Rankings.

| | Games | Avg. Ranking | | Gomoku | | Hold'em | | Bridge | | Chess | |
|---|---|---|---|---|---|---|---|---|---|---|---|
| | | Standard | Variant | Standard | Variant | Standard | Variant | Standard | Variant | Standard | Variant |
| Minimal | Claude-4-Sonnet | 3.75 | 3.75 | -0.096 | -0.075 | 0.001 | -0.023 | 0.005 | 0.128 | -0.042 | -0.075 |
| | Deepseek-Chat | 2.75 | 2.00 | 0.008 | 0.088 | 0.004 | 0.021 | 0.000 | 0.000 | 0.000 | 0.100 |
| | Doubao-Seed | 4.25 | 4.50 | 0.063 | -0.196 | -0.023 | -0.061 | -0.008 | -0.133 | -0.196 | 0.083 |
| | Gemini-2.5-Pro | 2.75 | 2.75 | 0.354 | 0.192 | 0.014 | 0.097 | -0.132 | -0.238 | -0.038 | 0.021 |
| | GPT-5 | 4.75 | 3.25 | 0.038 | -0.012 | -0.086 | 0.019 | -0.098 | 0.052 | -0.154 | -0.025 |
| | Qwen3-Coder | 2.75 | 4.75 | -0.092 | 0.029 | 0.000 | -0.080 | 0.037 | -0.008 | 0.025 | -0.167 |
| Commercial | best ADK | 2.50 | 3.00 | 0.260 | 0.000 | 0.018 | 0.039 | 0.127 | 0.238 | -0.094 | -0.104 |
| | Claude-Code | 3.25 | 1.88 | -0.042 | 0.083 | 0.091 | 0.194 | 0.060 | 0.081 | -0.104 | 0.047 |
| | CodeX | 3.50 | 3.50 | 0.104 | -0.089 | -0.034 | 0.033 | 0.023 | -0.090 | 0.010 | 0.052 |
| | Gemini-CLI | 2.75 | 4.25 | 0.120 | -0.021 | 0.031 | -0.083 | -0.027 | -0.169 | 0.188 | 0.010 |
| | Qwen-Coder | 3.00 | 2.38 | -0.130 | 0.193 | 0.041 | 0.077 | 0.092 | 0.081 | -0.062 | -0.021 |

As shown in Figure 5, we present the trends of global performance scores $G_i^n$ across four games, revealing distinct performance patterns for different models. In many cases, agents experience a sharp decline in performance during an intermediate round, which we interpret as a learning failure. Typically, such failures are recovered in the following round.

## E.2 DETAILED LEARNING SCORE

We list the detailed score of global learning, counter-adpatation learning and self-improvement in table 9, 10, 11 respectively.

The trend on four games are rather different. In general, the commercial model group consistently demonstrates superior global learning capability, where the advantage is particularly evident in complex strategy games like Chess or Gomoku variant.

Despite most minimal agents fail to learn well on complex games, we still find that Claude-4-Sonnet significantly surpass the rivals on standard games. However, the Claude-4-Sonnet still lacks behind on some cases like Gomoku, indicating that current LLMs agentic ablilities are still limited by the framework, where the commericial agents optimize workflow for their specific models to achieve the best results.

In simple games such as Hold'em, a larger proportion of agents exhibit positive learning scores, whereas in complex games like Chess, the prevalence of negative scores increases markedly. This trend suggests that current agents still face significant limitations in learning complex strategies.

Table 11: Self-improvement Score with Group-wise Average Rankings.

| Games | | Avg. Ranking | | Gomoku | | Hold'em | | Bridge | | Chess | |
|---|---|---|---|---|---|---|---|---|---|---|---|
| | | Standard | Variant | Standard | Variant | Standard | Variant | Standard | Variant | Standard | Variant |
| Minimal | Claude-4-Sonnet | 2.75 | 2.50 | -0.103 | -0.894 | 0.949 | 0.517 | 0.858 | 0.766 | -0.848 | 0.478 |
| | Deepseek-Chat | 3.25 | 3.38 | 0.000 | 0.893 | -0.949 | -0.747 | 0.000 | 0.000 | 0.000 | 0.000 |
| | Doubao-Seed | 3.75 | 4.38 | 0.141 | -0.686 | 0.075 | 0.000 | -0.202 | -0.775 | -0.894 | 0.000 |
| | Gemini-2.5-Pro | 3.63 | 2.00 | 0.400 | 0.775 | -0.758 | 0.894 | -0.598 | -0.240 | -0.775 | 0.913 |
| | GPT-5 | 4.88 | 4.50 | -0.897 | -0.949 | -0.205 | -0.050 | -0.767 | 0.485 | -0.775 | -0.390 |
| | Qwen3-Coder | 2.75 | 4.25 | -0.400 | 0.161 | 0.668 | 0.202 | 0.473 | -0.400 | -0.258 | -0.775 |
| Commercial | best ADK | 2.25 | 3.50 | 0.400 | 0.956 | 0.835 | -0.614 | 0.738 | 0.546 | -0.207 | -0.726 |
| | Claude-Code | 4.00 | 2.25 | -0.230 | 0.969 | 0.346 | 0.904 | 0.113 | 0.537 | -0.730 | -0.225 |
| | CodeX | 2.00 | 3.25 | 0.763 | -0.763 | -0.090 | 0.602 | 0.784 | 0.316 | 0.424 | 0.316 |
| | Gemini-CLI | 3.75 | 4.00 | -0.183 | -0.356 | -0.176 | 0.000 | 0.000 | -0.995 | 0.811 | -0.193 |
| | Qwen-Coder | 3.00 | 2.00 | 0.632 | 0.717 | 0.826 | 0.705 | 0.641 | 0.677 | -0.944 | 0.000 |

### E.3 BEHAVIORAL CHANGES INDUCED BY LEARNING

For each game, we randomly select 80-100 intermediate states from the agents' rival history and require the agent or LLM-player to choose the next move for each state. To ensure clarity in our writing, we uniformly refer to these intermediate states as *endgame* throughout the paper. Please note that *endgame* here is not limited to the final stages of the game; samples are taken from early, middle, and late stages as well.

We visualize the action consistency among the first two rounds in four games's endgame in Figure 6.

From the matrix, we observe that, in general, agents tend to learn the strategies of other agents in the first round (lower left part vs. upper left part). Specifically, Doubao-Seed and DeepSeek-Chat simply copy Claude-4-Sonnet's strategy in Holdem. Additionally, the learning trend is more pronounced in simpler games (Holdem vs. Chess).

For simpler games like Holdem, agent strategies in the second round are more similar to those in the first round (lower right part vs. upper left part), while for more difficult games like Chess, the trend is reversed. This observation is consistent with our findings on Trend$_{\text{mean}}$ in Table 5.

### E.4 CASE STUDY OF AGENT LEARNING

To demonstrate how agents actively improve their strategies by analyzing other agents' code and tournament reports, we present a detailed case study focusing on `Claude-Code_A` in Texas Hold'em Poker tournaments. This case clearly shows that score changes across rounds reflect the agent's understanding and adaptation to opponent strategies.

**Development Planning and Evidence of Learning**  Analysis of the Round 3 development log shows that the agent explicitly planned to study other agents' strategies. The planning log has following contexts:

```
1. "Read game project documentation and requirements" - completed
2. "Analyze tournament rules and mode" - ongoing
3. "Study last round tournament reports and AI strategies" - pending
4. "Examine existing AI code from round 2" - pending
```

**Tool Use Summary**  Table 12 summarizes all the learning-related tool use operations performed by `claude-code_A` during the development of its Round 3 code. The tool use logs show that the agent first located and read the tournament report file, extracted win statistics using `Grep` (codeX_A: 35 wins, gemini-cli_B: 30 wins, claude-code_A: 10 wins), then listed all agents, and finally read the code of the top performers.

**Strategy Adaptation in Code**  Comparing the code between rounds, it is evident that the agent made targeted improvements based on its analysis:

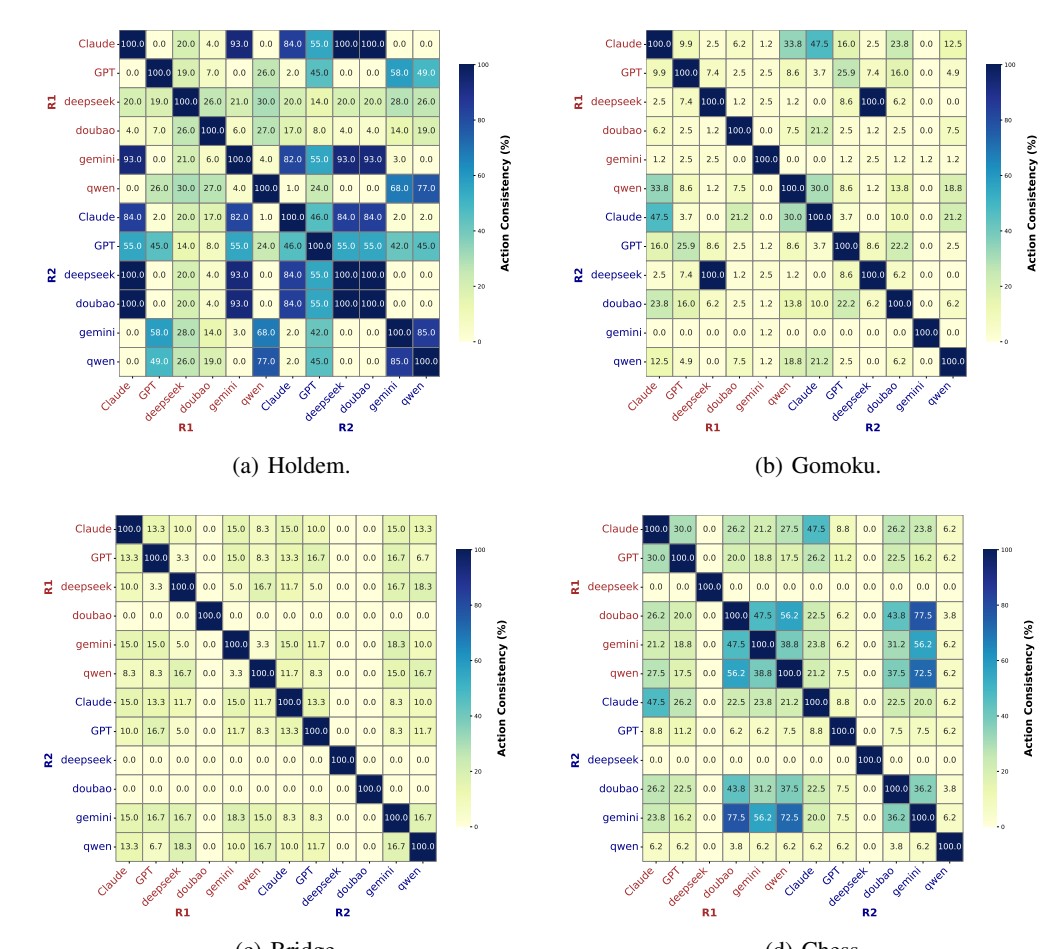

(a) Holdem.

(b) Gomoku.

(c) Bridge.

(d) Chess.

Figure 6: Action consistency between round 1 and round 2 agents' code on endgames.

Table 12: Summary of learning-related tool use operations by `claude-code_A` (Round 3 development).

| Tool | Operation | Purpose |
|------|-----------|---------|
| Glob | Search `**/tournament_report_tourney_*.json` | Locate tournament report files |
| Glob | Search `**/history_tourney_*.json` | Locate tournament history files |
| Read | Read `tournament_report_tourney_075455.json` | Analyze agent performance |
| Read | Read first 100 lines of report | Quickly understand report structure |
| Grep | Search for `"wins":` field | Extract win counts for each agent |
| Glob | List `round_2/**/*` | Discover all agents in previous round |
| Read | Read own previous code | Review existing implementation |
| Read | Read `codeX_A` code | Learn top-performing strategy (35 wins) |
| Read | Read `gemini-cli_B` code | Learn from another strong strategy (30 wins) |

- **Round 2:** Basic Chen formula evaluation, simple tournament strategy, no opponent modeling.

- **Round 3:** Added opponent analysis (`analyze_opponents()`), improved decision formula incorporating opponent aggression, refined stack and position factors.

Example of newly added code in Round 3:

```python
def analyze_opponents(self, game_state: Dict[str, Any]) -> Dict[str,
  float]:
    """Analyze opponent tendencies based on action history"""
```

```
action_history = game_state.get('action_history', [])
# ... analyze aggression for each opponent ...
return opponent_stats

# Use opponent analysis in decision making
opponent_stats = self.analyze_opponents(game_state)
avg_opponent_aggression = sum(opponent_stats.values()) /
↪    max(len(opponent_stats), 1)
decision_strength = (
    hand_strength *
    (1 + position_factor * 0.4) *
    stack_factor *
    bubble_factor *
    (1 + (1 - avg_opponent_aggression) * 0.2)
)
```

These code modifications are targeted responses to the strategies employed by top-performing agents, enabling more effective adaptation and strategic improvement.

**Reflections in Development Summary**    After completing development, the agent's summary explicitly compares its performance and highlights improvements:

---

Agent Self-Reflection Log

**Performance Analysis from Round 2**
Based on the tournament results:
- **codeX_A**: 35 wins (best performer)
- **gemini-cli_B**: 30 wins
- **claude-code_A**: 10 wins (my previous version)
My new AI improves on the previous version with:
- Better preflop ranges and position play
- Enhanced tournament strategy
- More sophisticated opponent modeling
- Improved risk management

---

Similar learning behavior is observed for `codeX_A` and other agents. This case study clearly demonstrates that agents actively analyze tournament reports and top-performing code via explicit tool use operations. The sequence and content of these operations, combined with targeted code improvements and reflective summaries, confirm that score changes across rounds are a reliable indicator of the agent's growing understanding and adaptation to competitor strategies.

## F    COMPARISON BETWEEN AGENT AND LLM-PLAYER

Table 13: Comparison of match outcomes between each agent and its corresponding LLM-Player. Each value indicates the agent's win rate when competing against the LLM that powers it.

| Agent VS LLM | Gomoku | | Hold'em | | Bridge | | Chess | |
|---|---|---|---|---|---|---|---|---|
| | Standard | Variant | Standard | Variant | Standard | Variant | Standard | Variant |
| Claude-4-Sonnet | 1.00 | 1.00 | 0.00 | 0.00 | 0.45 | 1.00 | 0.88 | 0.75 |
| Deepseek-Chat | 0.50 | 0.00 | 0.40 | 0.30 | 0.00 | 0.00 | 0.00 | 0.00 |
| Doubao-Seed | 0.50 | 0.25 | 0.00 | 0.00 | 0.00 | 0.10 | 0.00 | 0.00 |
| Gemini-2.5-Pro | 0.00 | 0.25 | 0.00 | 0.00 | 1.00 | 1.00 | 0.13 | 0.50 |
| GPT-5 | 0.00 | 0.50 | 0.00 | 0.50 | 1.00 | 0.00 | 0.00 | 0.00 |
| Qwen3-Coder | 1.00 | 0.00 | 0.50 | 0.50 | 1.00 | 0.00 | 0.50 | 0.50 |
| Claude-Code | 1.00 | 1.00 | 0.00 | 0.00 | 0.70 | 0.85 | 0.75 | 0.63 |
| CodeX | 0.00 | 0.50 | 0.20 | 0.40 | 1.00 | 0.38 | 1.00 | 1.00 |
| Gemini-CLI | 0.00 | 0.00 | 0.10 | 0.40 | 0.00 | 1.00 | 0.38 | 0.50 |
| Qwen-Coder | 1.00 | 1.00 | 0.50 | 0.00 | 1.00 | 0.00 | 0.56 | 0.63 |

We compare the strategies of agents' code and its corresponding LLM-Player on four games in table 13.

Interestingly, there is no strong correlation between the performance of the agents' code and that of their underlying models.

In games with strong strategic elements, such as Gomoku and Chess, some agents' code significantly outperforms their corresponding LLM-Player, indicating that the code implementation is able to better leverage game rules and strategies. For example, the agent developed by *claude-4-sonnet* achieves a 100% win rate against its LLM-Player in both standard and variant Gomoku, and also demonstrates a high win rate in Chess and Bridge. This suggests that the strategies implemented in the code are superior to the large model's direct reasoning performance as a player in these games. In contrast, the agents developed by *doubao-seed* and *deepseek-chat* struggle to defeat their respective models. However, in Hold'em, agents generally have lower win rates than the LLM-Player, possibly because the LLM-Player performs better in games with more psychological tactics, which are difficult to simulate with code while can be summarized by context learning.

We further visualize the action consistency between agents' code and LLM-palyers in Figure 7.

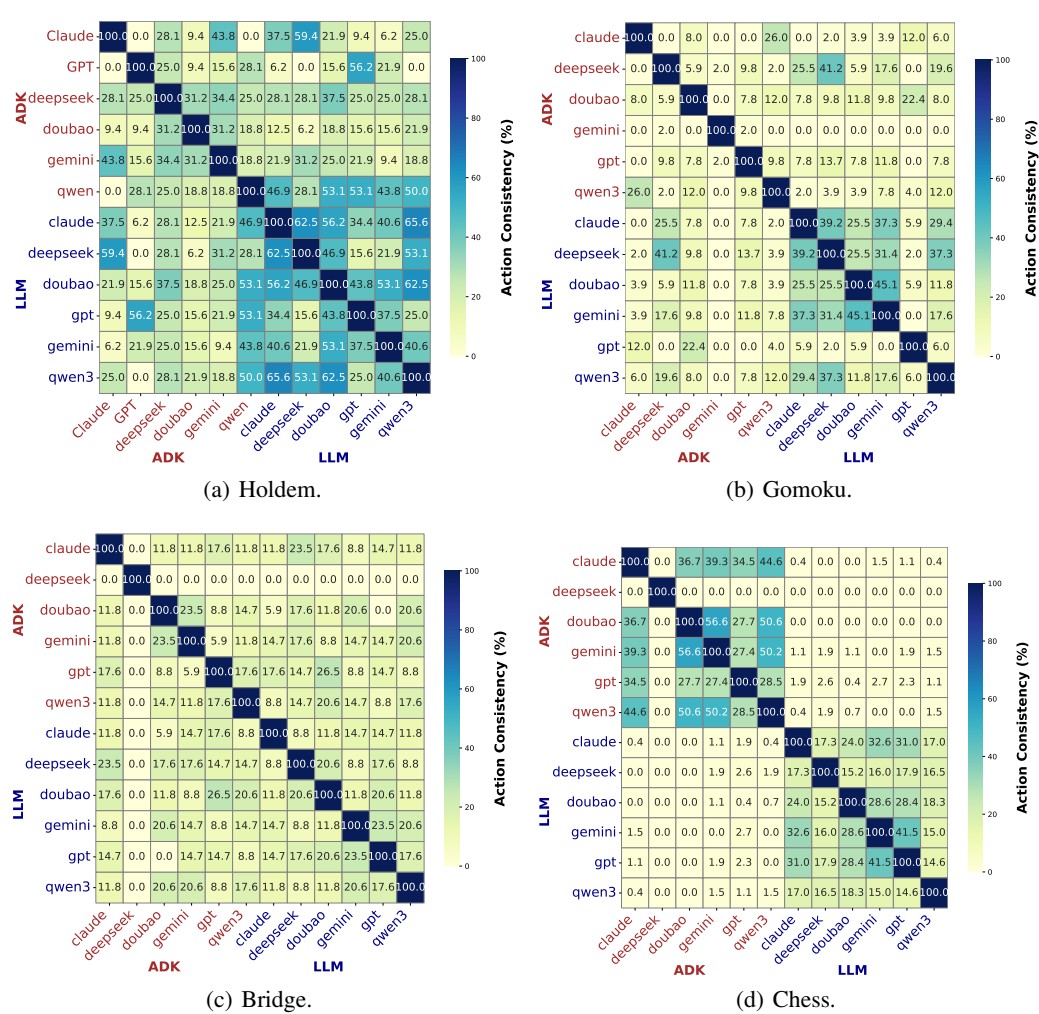

(a) Holdem.

(b) Gomoku.

(c) Bridge.

(d) Chess.

Figure 7: Action consistency between agents' code and LLM-Players on endgames.

We find that, in most games, the actions of LLM-players tend to resemble each other, and the strategies implemented in agents' code also exhibit high similarity among themselves. However, the code-based strategies and the plain reasoning of the same model often differ substantially. The only exception is Bridge, where we find numerous cases in which both LLM-players and code agents exhibit low consistency with human decisions. Considering that Bridge allows for a certain degree of decision freedom and its bidding rules are not strictly unified, we attribute this phenomenon to

the intrinsic characteristics of the game. Similar observations are also reported in other studies (Kita et al., 2024).

These findings further demonstrate that the strategies generated by the agents and those employed by the LLM-players are based on different approaches. This difference merits additional investigation in subsequent studies.

## G    CASE STUDY ON STRATEGIES

### G.1    STRATEGY COMPARISON ON CHESS

To highlight the fundamental differences between the LLM-player and traditional coding-based agents, we present a case study on move selection in a classical chess opening scenario. Consider the endgame visualized in Figure 8.

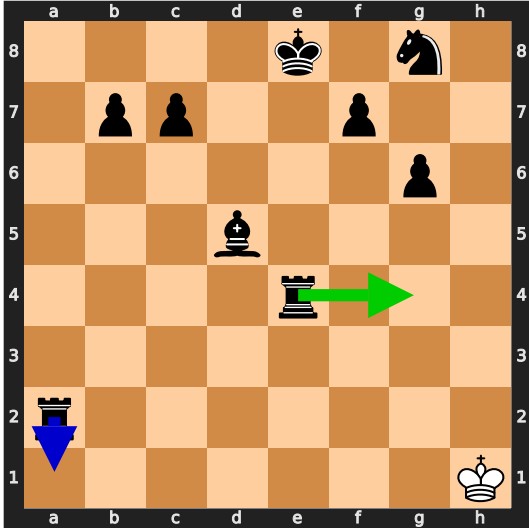

Figure 8: One end game situation of chess, black to move. The blue arrow indicates the move of GPT-5 LLM-player, and the green arrow indicates the move of GPT-5 Minimal code agents.

The GPT-5 LLM-player simulates chess reasoning by performing simple empirical analysis. When presented with a chess position, it first interprets the board state, assessing material balance and king safety: Black possesses two rooks, a bishop, and a knight, while White has only the king. The LLM observes that the White king is confined to the corner (h1) with no available escape squares, and immediately recalls the checkmate motif involving a rook on the first rank. Leveraging pattern recognition and accumulated chess knowledge, the LLM selects a2a1 as the optimal move to capture white king, closely mirroring human beginners' thought processes. This empirical process lead to sub-optimal move. The detailed CoT of the LLM-player is presented in the colorbox below.

> **Summarized Chain-of-Thought of LLM-player (GPT-5) for the end game**
>
> Let's break down the position from the FEN:
> **FEN:** `4 . . .`
> **Black to move.**
> **Board Analysis**
>
> - **Material:** Black is up a rook (two rooks vs. none), and has a bishop and knight. White has only a king.
> - **King Safety:** White's king is on h1, boxed in the corner. Black's king is on e8, safe.
> - **Threats:** Black has overwhelming material and can checkmate soon.
> - **Pieces:**
>   - Black rooks: a2, e4
>   - Black bishop: d5
>   - Black knight: g8
>   - Black pawns: c7, c8, f7, g6
>   - White king: h1
>
> **Tactical Considerations**
>
> - The White king is trapped on h1, with no escape squares.
> - Black can deliver checkmate if the king is attacked on the h-file or the second rank.
> - The rook on a2 can move to a1 for checkmate, as the king cannot escape.
>
> **Move Candidates**
>
> - **a2a1:** Rook moves to a1, delivering checkmate. The king on h1 cannot move to any square, and no piece can block or capture the rook.
> - Other moves (e.g., rook checks, bishop moves) are unnecessary, as a2a1 is immediate mate.
>
> **Reasoning**
>
> - **a2a1** is the best move because it delivers checkmate. The White king is trapped in the corner, and no piece can block or capture the rook. This ends the game immediately.
>
> —————————————————————————————————————————
>
> **Best Move**: `a2a1`

Conversely, the code agent thinks through numerical computation, relying on algorithmic search rather than empirical estimation. It generates all possible legal moves and evaluates them using a search tree, typically constrained by a fixed depth. In this case, the agent's search is limited to a fixed depth (5 plies), and its algorithm detect if the search reaches a terminal node where the king is actually checkmated. By search, the algorithm ensures e4g4 makes a checkmate and lead to win directly. The core code implementation of the LLM agent is shown in the Listing 1.

This divergence underscores that their reasoning mechanisms are not merely different implementations of the same algorithm, but represent distinct computational paradigms in chess problem solving.

## G.2 STRATEGY OF AGENTS

In Gomoku, strategies display clear stratification. Gemini and DeepSeek rely on random or near-random moves, while Claude, Doubao, GPT-5, and Qwen3 employ similar pattern-based evaluation with candidate filtering and Minimax search. Differences mainly lie in threat recognition, search control, and opening play: Claude and Doubao handle openings and forced moves more effectively, GPT-5 is steadier under time limits, Qwen3 remains balanced, whereas Gemini and DeepSeek are notably weaker.

The code similarity among agents in Texas Hold'em is relatively high, only DeepSeek employs a fully random strategy, while other models calculate winning probabilities based on the hand. On one

```
1350  def evaluate(board):
1351      """Evaluate the board position by summing material values for both
1352      ↪   sides."""
1353      piece_vals = {chess.PAWN: 100, chess.KNIGHT: 320, chess.BISHOP: 330,
1354                       chess.ROOK: 500, chess.QUEEN: 900, chess.KING: 0}
1355      score = 0
1356      for piece_type in piece_vals:
1357          score += piece_vals[piece_type] * len(board.pieces(piece_type,
1358          ↪   chess.WHITE))
1359          score -= piece_vals[piece_type] * len(board.pieces(piece_type,
1360          ↪   chess.BLACK))
1361      return score
1362
1363  def minimax(board, depth, maximizing):
1364      """Perform minimax search up to a fixed depth, recursively evaluating
1365      ↪   positions."""
1366      if depth == 0 or board.is_game_over():
1367          return evaluate(board)
1368      if maximizing:
1369          max_eval = -float('inf')
1370          for move in board.legal_moves:
1371              board.push(move)
1372              eval = minimax(board, depth - 1, False)
1373              board.pop()
1374              max_eval = max(max_eval, eval)
1375          return max_eval
1376      else:
1377          min_eval = float('inf')
1378          for move in board.legal_moves:
1379              board.push(move)
1380              eval = minimax(board, depth - 1, True)
1381              board.pop()
1382              min_eval = min(min_eval, eval)
1383          return min_eval
1384
1385  def find_best_move(fen, max_depth=5):
1386      """Given a FEN string, search for the best move using minimax to the
1387      ↪   given depth."""
1388      board = chess.Board(fen)
1389      best_move = None
1390      best_score = -float('inf')
1391      for move in board.legal_moves:
1392          board.push(move)
1393          score = minimax(board, max_depth - 1, False)
1394          board.pop()
1395          if score > best_score:
1396              best_score = score
1397              best_move = move
1398      return best_move
```

Listing 1: Core search and evaluation logic of GPT5 code agent generated by LLM Agent.

hand, this is because the available actions in Texas Hold'em are limited to fold, call/raise, and check. On the other hand, the strategies for Texas Hold'em are relatively straightforward to implement, as both reasoning and code are primarily based on hand strength. As a result, the code can closely simulate the reasoning process.

In the case of Chess, DeepSeek relies on an external library (Stockfish), but fails to configure it correctly, resulting in unsuccessful development. Even after multiple development iterations, DeepSeek continues to use this library without resolving the configuration issues. We also find that Claude, Doubao, Gemini, GPT, and Qwen3 utilize a similar combination of heuristic piece and board evaluation, Minimax search, and alpha-beta pruning, which leads to similar behavior. There are slight

differences in how each model evaluates the value of Chess pieces and the actions in endgame scenarios. Notably, Claude incorporates an opening book, which distinguishes it from the others and leads to better performance.

For Bridge, bidding and play strategies also stratify clearly. Qwen3 and Gemini rely on minimal logic, following random choices or only basic rules on High Card Points (HCP). By contrast, GPT-5, Doubao, and Claude incorporate structured evaluation, moving from total point counting (GPT-5) to multi-layered systems with suit quality, competitive actions, and signaling (Doubao and Claude). Despite these differences, all models share reliance on HCP as a core metric. Overall, Claude achieves the most complete integration of evaluation and play, Doubao is comparably advanced, GPT-5 remains simpler, while Gemini and Qwen3 lag behind.

## H EXTENDED ROUND ANALYSIS

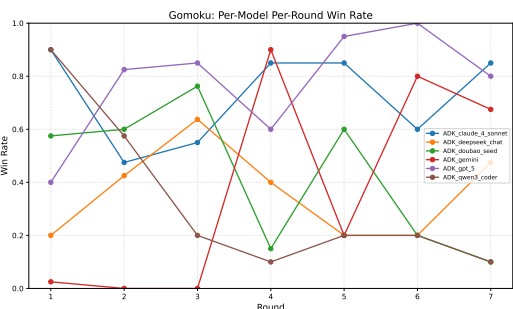

Figure 9: $G_i^n$ Trend of Minimal Agent in standard Gomoku game across 7 rounds.

We conduct additional experiments on the standard Gomoku task with seven learning rounds to analyze the dynamics of agent performance over time. As shown in Figure 9, the $G_i^n$ trend of minimal agents exhibits significant changes during the first four rounds, indicating an active learning phase with substantial strategy refinement and exploration. We calculate the ranking variance among agents in Round 1-4 is 1.47, while the ranking variance among agents in Round 4-7 decreases to 0.75, demonstrating that agent performance becomes more stable as learning progresses. These results confirm that the early rounds are critical for capturing the agents' learning and adaptation behaviors, while later rounds reflect a stabilization in performance. Therefore, using the first four rounds for evaluation provides a reliable and representative assessment of learning ability, and our metrics remain robust throughout different stages of agent development.

## I RESULTS OF ML TRACK

Table 14: ML ability scores and average rankings of agents.

| | Agent | Gomoku↑ | Hold'em↑ | Bridge↑ | Chess↑ | Avg. Ranking↓ |
|---|---|---|---|---|---|---|
| Minimal | Claude-4-Sonnet | **0.787** | 0.360 | 0.600 | 0.700 | 2.25 |
| | DeepSeek-Chat | 0.612 | 0.000 | 0.170 | 0.100 | 4.25 |
| | Doubao-Seed | 0.375 | 0.110 | 0.290 | 0.100 | 4.25 |
| | Gemini-2.5-Pro | 0.000 | 0.000 | 0.140 | 0.675 | 5.00 |
| | GPT-5 | 0.625 | **0.530** | **0.900** | 0.700 | **1.50** |
| | Qwen3-Coder | 0.600 | 0.000 | **0.900** | **0.725** | 2.50 |
| Commercial | best ADK | **0.750** | 0.190 | **0.700** | **0.656** | **1.25** |
| | Claude-Code | 0.578 | 0.170 | 0.000 | 0.406 | 4.00 |
| | CodeX | 0.484 | 0.190 | 0.400 | 0.469 | 3.00 |
| | Gemini-CLI | 0.187 | **0.280** | 0.200 | 0.438 | 3.50 |
| | Qwen-Coder | 0.500 | 0.170 | **0.700** | 0.531 | 2.50 |

The detailed results of agents' performance on machine learning devloments is shown in table 14.

In ML track, agents autonomously generate data, design code, train models, and deliver ML-based strategies in a GPU-enabled environment. The results of ML ability is provided in Appendix I. Most

agents only implement basic models with limited training, resulting in smaller performance gaps and different rankings compared to the strategy track.

## J  RESULTS OF MULTI-LINGUAL TRACK

We list the detailed results on different languages in  15.

Table 15: Scores of agents on games with different languages, with variance analysis.

| Agent | | Gomoku | | | | Hold'em | | | | Bridge | | | | Avg. Variance↓ |
|---|---|---|---|---|---|---|---|---|---|---|---|---|---|---|
| | | Python↑ | JS↑ | Go↑ | Var.↓ | Python↑ | JS↑ | Go↑ | Var.↓ | Python↑ | JS↑ | Go↑ | Var.↓ | |
| Minimal | Claude-4-Sonnet | 1.000 | 0.250 | 0.250 | 0.125 | 0.360 | 0.640 | 0.000 | 0.069 | 1.000 | 0.250 | 0.250 | 0.125 | 0.106 |
| | DeepSeek-Chat | 1.000 | 0.250 | 0.250 | 0.125 | 1.000 | 0.000 | 0.000 | 0.222 | 0.500 | 0.500 | 0.500 | 0.000 | 0.116 |
| | Doubao-Seed | 1.000 | 0.500 | 0.000 | 0.167 | 1.000 | 0.000 | 0.000 | 0.222 | 0.500 | 1.000 | 0.000 | 0.167 | 0.185 |
| | Gemini-2.5-Pro | 0.750 | 0.750 | 0.000 | 0.125 | 0.010 | 0.990 | 0.000 | 0.216 | 1.000 | 0.250 | 0.250 | 0.125 | 0.155 |
| | GPT-5 | 0.500 | 0.000 | 1.000 | 0.167 | 1.000 | 0.000 | 0.000 | 0.222 | 1.000 | 0.500 | 0.000 | 0.167 | 0.185 |
| | Qwen3-Coder | 1.000 | 0.250 | 0.250 | 0.125 | 0.610 | 0.290 | 0.100 | 0.044 | 0.688 | 0.000 | 0.812 | 0.128 | **0.099** |
| Commercial | Claude-Code | 1.000 | 0.250 | 0.250 | 0.125 | 0.200 | 0.020 | 0.780 | 0.105 | 1.000 | 0.250 | 0.250 | 0.125 | 0.118 |
| | CodeX | 0.000 | 0.500 | 1.000 | 0.167 | 0.200 | 0.030 | 0.770 | 0.100 | 1.000 | 0.000 | 0.500 | 0.167 | 0.145 |
| | Gemini-CLI | 0.750 | 0.750 | 0.000 | 0.125 | 1.000 | 0.000 | 0.000 | 0.222 | 1.000 | 0.250 | 0.250 | 0.125 | 0.157 |
| | Qwen-Coder | 1.000 | 0.250 | 0.250 | 0.125 | 1.000 | 0.000 | 0.000 | 0.222 | 0.975 | 0.000 | 0.525 | 0.159 | 0.169 |

Most agents achieve their highest scores in Python, while several models exhibit significant performance fluctuations in JS and Go. Qwen3-Coder demonstrates the most consistent results across all languages, with the lowest average variance, indicating strong cross-language adaptability. In contrast, models such as GPT-5 and Doubao-Seed show considerable differences between languages, reflecting limited generalization ability. Commercial agents also exhibit score differences across different programming languages. Considering that board game strategies are inherently language-agnostic tasks, the performance gaps observed in the multi-language track of CATArena indicate that current agents are not yet able to effectively abstract strategies into unified algorithms and implement them consistently across languages. Such algorithmic abstraction should be a key direction for the future development of agents.

## K  ERROR RATE ANALYSIS

During the implementation of strategy code, some agents encounter runtime errors. We observe that all agents were able to generate the required files successfully. However, most errors occur in interface implementation and action validity. These cases were consistently recorded as HTTP errors. Whenever an HTTP error occurs, a default action is automatically executed to ensure the continuity of the competition. For Texas Hold'em Poker, the default action is to fold for that hand. For chess and Gomoku, the default action is for the player to concede the current game.

Table 16: Proportion of HTTP errors encountered during code execution for each agent in Round 1.

| HTTP error(%) | | Gomoku | | Hold'em | | Bridge | | Chess | |
|---|---|---|---|---|---|---|---|---|---|
| | | Standard | Variant | Standard | Variant | Standard | Variant | Standard | Variant |
| Minimal | Claude-4-Sonnet | 7 | 0 | 0 | 0 | 0 | 0 | 0 | 0 |
| | Deepseek-Chat | 21 | 0 | 0 | 48 | 0 | 0 | 100 | 100 |
| | Doubao-Seed | 17 | 0 | 49 | 93 | 12 | 0 | 0 | 100 |
| | Gemini-2.5-Pro | 3 | 100 | 8 | 50 | 0 | 1 | 0 | 0 |
| | GPT-5 | 6 | 0 | 0 | 0 | 0 | 100 | 2 | 21 |
| | Qwen3-Coder | 11 | 0 | 0 | 91 | 2 | 1 | 0 | 0 |
| Commercial | best ADK | 0 | 0 | 0 | 0 | 0 | 0 | 0 | 0 |
| | Claude-Code | 0 | 0 | 0 | 0 | 0 | 0 | 0 | 0 |
| | CodeX | 0 | 0 | 0 | 0 | 0 | 0 | 0 | 0 |
| | Gemini-CLI | 0 | 0 | 0 | 0 | 48 | 0 | 0 | 0 |
| | Qwen-Coder | 2 | 0 | 0 | 18 | 0 | 17 | 0 | 0 |

Table 16 reports the proportion of HTTP error actions encountered during the code development of each agent in Round 1. The error rate is calculated as the rate of failed actions in total number of actions taken by the strategy code. The experimental results indicate that commercial agents exhibit a significantly lower failure rate compared to minimal agents. Additionally, the failure rate is higher in variant games than in traditional games.

## L    COST AND CODE COMPLEXITY OF PARTICIPANTS

We list the agents' cost and code statistics in Table 17 for first round development of standard games, 18 for second round development of standard games, 19 for first round development of variant games and  20 for second round development of variant games.

We can see that game development token costs show minimal variation, while differences is significant due to model changes. Claude (both minimal and code-based agents) consumes significantly more input tokens than competitors, exceeding the average by over 2 times, while Gemini generates notably more output tokens compared to other models. GPT-5 offers the best trade-off between cost. Among all agents, second-round game development require more input tokens, while output token growth remains marginal. In addition, commercial agents consistently use fewer tokens than their minimal-agent counterparts.

In terms of code complexity, agents driven by Claude-4 model consistently surpasses other agents in both the number of effective lines of code developed and the time spent considering development strategies. We observe that its development strategies are more sophisticated. Additionally, the complexity of its code increases with each iteration, which indirectly demonstrates the model's exceptional learning capabilities.

## M    FULL PROMPTS

The agent is instructed to develop a competitive game AI based on the provided game environment. The AI must be deployed as an HTTP service with a single-port startup script, follow the official development instructions, and be named with the model prefix. And the agent is encouraged to iteratively improve its strategy based on the tournament report and the previous codes. Full prompt of details are in Table 21 for main leaderboard,  22 for ML track and  23 for multilingual track.

Table 22:  Machine Learning Game AI with MANDATORY Self-Play Training Prompt

---

**Machine Learning Game AI with MANDATORY Self-Play Training**

---

Develop a competitive game AI for *game env path* using **REAL machine learning with actual training**.
**CRITICAL: NO PSEUDO-ML ALLOWED**
**MANDATORY:** Implement real training with actual parameter updates.

**Forbidden:** Random weights, unused optimizers, no training loops
**Required:** Self-play training, `loss.backward()`, `optimizer.step()`, saved trained model
**Training Requirements**
1. **Self-Play System:** Generate training data by playing against itself
2. **Training Loop:** Real parameter updates with backpropagation
3. **Model Saving:** Save trained model weights (e.g., `trained_model.pth`)
4. **Training Endpoint:** `/train` HTTP endpoint to trigger training

**Technical Implementation**
The final AI should be provided as an HTTP service. You can refer to the guides in *game env path*/README.md and *game env path*/develop_instruction.md for development instructions. **The content in *game env path*/develop_instruction.md is very important, please read it carefully!**
Please develop your AI service directly under *dir path*.

**Script Requirements**
Implement a script to start your AI service, with the name `start_ai.sh` in *dir path*. The script must accept exactly one argument, the port number to run the HTTP service. Start the AI via:
`bash start_ai.sh <port>`

---

| Table 22 – continued from previous page |
| --- |
| **Machine Learning Game AI with MANDATORY Self-Play Training** |

Check service health via: `curl -s http://localhost:<port>/health`

**Additional Training Script:** Also create `train_ai.sh` for self-play training:
`bash train_ai.sh <num_episodes>`

**Note:** The script should not accept any other arguments except for the port number. Ensure the AI service uses this port for HTTP requests.

**Environment Requirements**

**MANDATORY:** You MUST develop under the `ml_env_py311` conda environment. PyTorch, TensorFlow, NumPy, pandas, and scikit-learn are already installed.

- DO NOT include PyTorch or TensorFlow in `requirements.txt`
- Create a requirements.txt file only for additional dependencies you need beyond the pre-installed packages
- All development and execution must be done in the `ml_env_py311` environment

**Available Game Port**

We offer an interactive game port: `http://localhost:game port` to facilitate your testing and training. You can interact with the service and get feedback from the port.

**Validation Requirements**

Submission will be rejected if:
- The model uses only random weights without training
- No actual training loop with parameter updates
- No self-play data generation system
- Cannot demonstrate learning/improvement over time
- Training endpoints return fake responses without real training

**Other Requirements**

Use your model name as a prefix in the name of your AI service, i.e., *model name*_AI.

Develop directly in *dir path* without repeatedly asking for the next step. Report to me only after you have completed the development.

**Final Remind**

You should write `start_ai.sh` and `train_ai.sh` in *dir path* and implement the REAL ML-based AI service with actual training capabilities in *dir path*. DO NOT MODIFY THE CODE IN *game env path*. Please make sure to implement a genuine machine learning training workflow, including writing the code, performing training, tuning hyperparameters, and testing the model. After training, save the trained model parameters, and ensure that the AI service can be started with the specified script to perform inference.

**Mandatory:** Submission must include a trained model file (e.g., `trained_model.pth`, `model_weights.pkl`) proving real training.

**Zero Tolerance for Pseudo-ML:** Any submission without real training will be rejected.

Table 17: Comparison of model cost on four games of 1st round.

| Metric | Claude-4-Sonnet | Deepseek-Chat | Doubao-Seed | Gemini-2.5-Pro | GPT-5 | Qwen3-Coder | Claude-Code | CodeX | Gemini-CLI | Qwen-Coder |
|---|---|---|---|---|---|---|---|---|---|---|
| **Gomoku** | | | | | | | | | | |
| Total input tokens | 825964 | 34266 | 83307 | 1074278 | 392378 | 251590 | 925166 | - | 169567 | 623393 |
| Session tokens | 31571 | 8256 | 20186 | 27481 | 25201 | 20013 | 29340 | - | 21590 | 26478 |
| Total output tokens | 13777 | 1400 | 9432 | 25952 | 14184 | 6338 | 11746 | - | 2303 | 8002 |
| Total time (s) | 441.916 | 29.021 | 206.159 | 858.027 | 280.399 | 135.770 | 323.200 | - | 519.100 | 607.200 |
| Tools used | 68 | 12 | 20 | 118 | 44 | 32 | 38 | 24 | 8 | 26 |
| Valid lines of code | 574 | 69 | 405 | 65 | 421 | 425 | 350 | 347 | 101 | 335 |
| Avg thinking time (s) | 0.0337 | 0.0026 | 2.9848 | 0.0026 | 1.9681 | 0.0903 | 0.0053 | 0.7026 | 0.1151 | 0.0036 |
| **Texas Hold'em Poker** | | | | | | | | | | |
| Total input tokens | 966609.5 | 44883 | 97274 | 168565 | 345444 | 326508 | 296329.5 | - | 840690.5 | 1268906.5 |
| Session tokens | 45384 | 9388 | 14413 | 14073 | 26418 | 16688 | 29663.5 | - | 27828 | 23646.5 |
| Total output tokens | 20974 | 985 | 10370 | 40762 | 24178 | 4472 | 8537 | - | 11833.5 | 12772 |
| Total time (s) | 489.150 | 25.385 | 256.145 | 481.315 | 473.910 | 131.630 | 143.420 | - | 365.485 | 474.075 |
| Tools used | 38 | 8 | 12 | 15 | 20 | 24 | 12 | 19 | 39 | 48 |
| Valid lines of code | 742 | 21 | 169 | 92 | 346 | 164 | 122 | 318 | 300 | 220 |
| Avg thinking time (s) | 0.0026 | 0.0023 | 0.0030 | 0.0025 | 0.1004 | 0.0025 | 0.0028 | 0.0021 | 0.0024 | 0.0023 |
| **Bridge** | | | | | | | | | | |
| Total input tokens | 1429350 | 18027 | 81206 | 266506 | 258633 | 114679 | 618991 | - | 204913 | 71696 |
| Session tokens | 38087 | 15077 | 24374 | 29486 | 35448 | 20358 | 34263 | - | 46945 | 34937 |
| Total output tokens | 17039 | 1131 | 19247 | 12713 | 3954 | 5394 | 12284 | - | 13222 | 20308 |
| Total time (s) | 727.550 | 3.523 | 333.361 | 191.775 | 128.895 | 334.394 | 274.200 | - | 188.700 | 613.000 |
| Tools used | 94 | 6 | 14 | 32 | 26 | 16 | 25 | 24 | 5 | 32 |
| Valid lines of code | 521 | 0 | 552 | 280 | 245 | 271 | 464 | 362 | 349 | 549 |
| Avg thinking time (s) | 0.0095 | 0.0000 | 0.0105 | 0.0094 | 0.0095 | 0.0093 | 0.0091 | 0.0091 | 0.0093 | 0.0093 |
| **Chess** | | | | | | | | | | |
| Total input tokens | 1612835 | 49655 | 86674 | 346890 | 484611 | 267156 | 709737 | - | 153825 | 23489 |
| Session tokens | 38729 | 9953 | 14799 | 93292 | 39962 | 20781 | 29818 | - | 25418 | 23620 |
| Total output tokens | 16584 | 1257 | 14411 | 80997 | 28695 | 6624 | 13017 | - | 4727 | 6845 |
| Total time (s) | 617.339 | 26.882 | 287.976 | 356.551 | 444.932 | 141.417 | 285.593 | - | 162.342 | 142.344 |
| Tools used | 110 | 14 | 16 | 40 | 44 | 34 | 32 | 14 | 6 | 18 |
| Valid lines of code | 734 | 0 | 292 | 359 | 437 | 360 | 393 | 283 | 262 | 325 |
| Avg thinking time (s) | 1.3500 | 0.0000 | 0.0030 | 0.0030 | 0.2690 | 0.0020 | 1.9730 | 0.0030 | 0.0030 | 0.0050 |
| **Average across games** | | | | | | | | | | |
| Total input tokens | 1208689.625 | 36707.75 | 87115.25 | 464059.75 | 370266.5 | 239983.25 | 637555.875 | - | 342248.875 | 496871.125 |
| Session tokens | 38442.75 | 10668.5 | 18443.0 | 41083.0 | 31757.25 | 19460.0 | 30771.125 | - | 30445.25 | 27170.375 |
| Total output tokens | 17093.5 | 1193.25 | 13365.0 | 40106.0 | 17752.75 | 5707.0 | 11396.0 | - | 8021.375 | 11981.75 |
| Total time (s) | 568.989 | 21.203 | 270.910 | 471.917 | 332.034 | 185.803 | 256.603 | - | 308.907 | 459.155 |
| Tools used | 77.5 | 10.0 | 15.5 | 51.25 | 33.5 | 26.5 | 26.75 | 20.25 | 14.375 | 30.875 |
| Valid lines of code | 642.75 | 22.5 | 354.5 | 199.0 | 362.25 | 305.0 | 332.25 | 327.5 | 253.0 | 357.25 |
| Avg thinking time (s) | 0.3489 | 0.0012 | 0.7503 | 0.0044 | 0.5867 | 0.0260 | 0.4975 | 0.1792 | 0.0324 | 0.0050 |

Table 18: Comparison of model cost on four games of 2nd round.

| Metric | Claude-4-Sonnet | Deepseek-Chat | Doubao-Seed | Gemini-2.5-Pro | GPT-5 | Qwen3-Coder | Claude-Code | CodeX | Gemini-CLI | Qwen-Coder |
|---|---|---|---|---|---|---|---|---|---|---|
| **Gomoku** | | | | | | | | | | |
| Total input tokens | 1924785 | 170294 | 625384 | 589934 | 1351710 | 4859091 | 761468 | - | 514517 | 573759 |
| Session tokens | 102767 | 3495 | 84577 | 38760 | 78092 | 151664 | 37085 | - | 53402 | 27637 |
| Total output tokens | 14154 | 1608 | 16828 | 17098 | 10126 | 8230 | 11705 | - | 5180 | 7006 |
| Total time (s) | 433.112 | 765.076 | 436.142 | 1195.109 | 387.641 | 918.349 | 265.7 | - | 104.4 | 273.5 |
| Tools used | 58 | 62 | 32 | 58 | 48 | 82 | 30 | 22 | 12 | 23 |
| Valid lines of code | 617 | 69 | 527 | 305 | 467 | 368 | 493 | 302 | 251 | 376 |
| Avg thinking time (s) | 1.9630 | 0.0026 | 2.1291 | 0.0000 | 1.5888 | 0.3499 | 0.7542 | 0.2172 | 0.0080 | 0.0047 |
| **Texas Hold'em Poker** | | | | | | | | | | |
| Total input tokens | 1497923 | 56126 | 225806 | 2076630 | 1328520 | 3671952 | 1143571 | - | 1048050 | 1071708 |
| Session tokens | 143575 | 9819 | 57890 | 240057 | 113649 | 323278 | 53929 | - | 40566 | 49161 |
| Total output tokens | 7309 | 1151.5 | 10263.5 | 46280.5 | 13269.5 | 5986.5 | 7349.5 | - | 12617 | 6954 |
| Total time (s) | 308.05 | 45.935 | 327.225 | 1171.58 | 691.74 | 1057.84 | 866.715 | - | 1453.08 | 163.275 |
| Tools used | 16 | 11 | 10 | 17 | 21 | 25 | 28 | 18 | 41 | 27 |
| Valid lines of code | 283 | 21 | 110 | 156 | 362 | 293 | 264 | 319 | 363 | 315 |
| Avg thinking time (s) | 0.0026 | 0.0024 | 0.0024 | 0.0019 | 0.0912 | 0.0026 | 0.0023 | 0.0024 | 0.0025 | 0.0025 |
| **Bridge** | | | | | | | | | | |
| Total input tokens | 740733 | 18388 | 184814 | 277299 | 900865 | 720015 | 940709 | - | 275340 | 105056 |
| Session tokens | 28130 | 15318 | 4076 | 131089 | 119593 | 62734 | 38388 | - | 49946 | 63756 |
| Total output tokens | 14200 | 1178 | 18169 | 7520 | 4983 | 10270 | 15530 | - | 8161 | 25125 |
| Total time (s) | 397.164 | 6.097 | 391.047 | 162.147 | 367.204 | 1863.073 | 365.9 | - | 169.0 | 1337.7 |
| Tools used | 48 | 8 | 28 | 26 | 68 | 44 | 33 | 17 | 6 | 38 |
| Valid lines of code | 650 | 0 | 731 | 271 | 245 | 288 | 700 | 367 | 280 | 1112 |
| Avg thinking time (s) | 0.0101 | 0.0000 | 0.0135 | 0.0099 | 0.0096 | 0.0099 | 0.0105 | 0.0105 | 0.0102 | 0.0106 |
| **Chess** | | | | | | | | | | |
| Total input tokens | 2118018 | 45051 | 154263 | 426659 | 390211 | 1014853 | 4129953 | - | 308116 | 31752 |
| Session tokens | 101195 | 3954 | 28503 | 26538 | 35482 | 36705 | 72658 | - | 33919 | 31877 |
| Total output tokens | 13067 | 412 | 10063 | 13435 | 16798 | 11018 | 25438 | - | 6283 | 7250 |
| Total time (s) | 1352.194 | 522.132 | 230.664 | 931.231 | 214.792 | 263.665 | 906.521 | - | 420.146 | 177.453 |
| Tools used | 44 | 26 | 28 | 44 | 34 | 80 | 78 | 14 | 13 | 28 |
| Valid lines of code | 647 | 0 | 305 | 347 | 559 | 499 | 736 | 351 | 299 | 335 |
| Avg thinking time (s) | 0.8240 | 0.0000 | 0.0020 | 0.0000 | 0.0030 | 0.0020 | 0.0030 | 0.0040 | 1.8470 | 0.0050 |
| **Average across games** | | | | | | | | | | |
| Total input tokens | 1570364.75 | 72464.75 | 297566.75 | 842630.5 | 992826.5 | 2566477.75 | 1743925.25 | - | 536505.75 | 445568.75 |
| Session tokens | 93916.75 | 8146.5 | 43761.5 | 109111.0 | 86704.0 | 143595.25 | 50515.0 | - | 44458.25 | 43107.75 |
| Total output tokens | 12182.5 | 1087.38 | 13830.88 | 21083.38 | 11294.13 | 8876.13 | 15005.63 | - | 8060.25 | 11583.75 |
| Total time (s) | 622.63 | 334.81 | 346.27 | 865.02 | 415.34 | 1025.73 | 601.21 | - | 536.66 | 487.98 |
| Tools used | 41.5 | 26.75 | 24.5 | 36.25 | 42.75 | 57.75 | 42.25 | 17.75 | 18.0 | 29.0 |
| Valid lines of code | 549.25 | 22.5 | 418.25 | 269.75 | 408.25 | 362.0 | 548.25 | 334.75 | 298.25 | 534.5 |
| Avg thinking time (s) | 0.7000 | 0.0013 | 0.5367 | 0.0029 | 0.4231 | 0.0911 | 0.1925 | 0.0585 | 0.4669 | 0.0057 |

Table 19: Comparison of model cost on four variant games of 1st round.

| Metric | Claude-4-Sonnet | Deepseek-Chat | Doubao-Seed | Gemini-2.5-Pro | GPT-5 | Qwen3-Coder | Claude-Code | CodeX | Gemini-CLI | Qwen-Coder |
|---|---|---|---|---|---|---|---|---|---|---|
| **Gomoku** | | | | | | | | | | |
| Total input tokens | 833531 | 35489 | 83792 | 342027 | 539388 | 68087 | 495982 | - | 256919 | 23991 |
| Complete tokens | 22521 | 8587 | 23512 | 27175 | 27291 | 11468 | 27275 | - | 39232 | 24082 |
| Total output tokens | 10900 | 1394 | 11740 | 13553 | 19676 | 1658 | 8153 | - | 3522 | 7436 |
| Total time (s) | 307.319 | 24.007 | 255.817 | 809.100 | 365.077 | 73.260 | 191.800 | - | 74.300 | 140.200 |
| Tools used | 64 | 14 | 20 | 36 | 52 | 14 | 23 | 13 | 7 | 18 |
| Valid lines of code | 859 | 64 | 746 | 414 | 301 | 298 | 254 | 313 | 156 | 431 |
| Avg thinking time (s) | 0.0473 | 0.0026 | 0.0333 | 6.7802 | 0.0904 | 0.0048 | 0.0106 | 0.0051 | 0.0124 | 0.7938 |
| **Texas Hold'em Poker** | | | | | | | | | | |
| Total input tokens | 1182600 | 39782 | 89270 | 2582120 | 191064 | 385246 | 160410 | - | 864568 | 775880 |
| Complete tokens | 39575 | 9174 | 14448 | 32547 | 16530 | 16013 | 24287 | - | 25654 | 25308 |
| Total output tokens | 13110 | 962 | 9897 | 52457 | 16407 | 4361 | 5801 | - | 10699 | 7568 |
| Total time (s) | 398.130 | 20.195 | 217.600 | 1337.345 | 558.055 | 124.640 | 84.920 | - | 319.745 | 270.315 |
| Tools used | 42 | 7 | 11 | 86 | 17 | 30 | 7 | 19 | 42 | 34 |
| Valid lines of code | 469 | 21 | 204 | 98 | 302 | 144 | 167 | 256 | 232 | 243 |
| Avg thinking time (s) | 0.0026 | 0.0022 | 0.0011 | 0.0486 | 0.0019 | 0.0011 | 0.0025 | 0.0801 | 0.0028 | 0.0021 |
| **Bridge** | | | | | | | | | | |
| Total input tokens | 475377 | 16805 | 122427 | 232284 | 110768 | 653245 | 1908005 | - | 173628 | 26896 |
| Complete tokens | 37640 | 13764 | 36846 | 22593 | 4433 | 4438 | 53465 | - | 33184 | 27023 |
| Total output tokens | 16704 | 1137 | 7583 | 1020 | 2378 | 22823 | 18489 | - | 7412 | 5191 |
| Total time (s) | 298.800 | 7.721 | 158.254 | 483.948 | 304.010 | 7111.237 | 462.300 | - | 229.200 | 129.600 |
| Tools used | 34 | 6 | 16 | 32 | 40 | 308 | 55 | 16 | 5 | 22 |
| Valid lines of code | 627 | 0 | 291 | 213 | 133 | 165 | 469 | 368 | 212 | 224 |
| Avg thinking time (s) | 0.0071 | 0.0000 | 0.0068 | 0.0068 | 0.0096 | 0.0067 | 0.0089 | 0.0089 | 0.0060 | 0.0092 |
| **Chess** | | | | | | | | | | |
| Total input tokens | 1000994 | 38977 | 84149 | 432950 | 229279 | 253674 | 979611 | - | 184178 | 115022 |
| Complete tokens | 38485 | 7761 | 17321 | 43002 | 20520 | 17592 | 34597 | - | 26984 | 19923 |
| Total output tokens | 16355 | 1125 | 10518 | 32332 | 16499 | 5948 | 13010 | - | 6014 | 8032 |
| Total time (s) | 473.286 | 23.186 | 241.221 | 729.121 | 283.023 | 144.435 | 357.587 | - | 139.114 | 368.952 |
| Tools used | 74 | 14 | 14 | 40 | 32 | 38 | 41 | 12 | 8 | 19 |
| Valid lines of code | 719 | 41 | 399 | 398 | 475 | 343 | 410 | 427 | 152 | 392 |
| Avg thinking time (s) | 0.7520 | 0.0000 | 0.0000 | 0.2560 | 1.1190 | 0.0020 | 1.9660 | 0.0040 | 0.0040 | 0.0050 |
| **Average across games** | | | | | | | | | | |
| Total input tokens | 873125.5 | 32763.25 | 94909.5 | 897345.25 | 267624.75 | 340063.0 | 886002.0 | - | 369823.25 | 235447.25 |
| Complete tokens | 34555.25 | 9821.5 | 23031.75 | 31329.25 | 17193.5 | 12377.75 | 34906.0 | - | 31263.5 | 24084.0 |
| Total output tokens | 14267.25 | 1154.5 | 9934.5 | 27090.5 | 13740.0 | 8697.5 | 11363.25 | - | 6911.75 | 7056.75 |
| Total time (s) | 369.3838 | 18.7773 | 218.2230 | 839.8785 | 377.5413 | 1863.3930 | 274.1518 | - | 190.5898 | 227.2668 |
| Tools used | 53.5 | 10.25 | 15.25 | 48.5 | 35.25 | 97.5 | 31.5 | 15.0 | 15.5 | 23.25 |
| Valid lines of code | 668.5 | 31.5 | 410.0 | 280.75 | 302.75 | 237.5 | 325.0 | 341.0 | 188.0 | 322.5 |
| Avg thinking time (s) | 0.2022 | 0.0012 | 0.0103 | 1.7729 | 0.3052 | 0.0036 | 0.4970 | 0.0245 | 0.0063 | 0.2025 |

Table 20: Comparison of model cost on four variant games of 2nd round.

| Metric | Claude-4-Sonnet | Deepseek-Chat | Doubao-Seed | Gemini-2.5-Pro | GPT-5 | Qwen3-Coder | Claude-Code | CodeX | Gemini-CLI | Qwen-Coder |
|---|---|---|---|---|---|---|---|---|---|---|
| **Gomoku** | | | | | | | | | | |
| Total input tokens | 1298765 | 111940 | 680676 | 545504 | 606704 | 1542046 | 805252 | - | 227722 | 873934 |
| Complete tokens | 132219 | 1100 | 121218 | 24003 | 82812 | 4309 | 38629 | - | 41093 | 32599 |
| Total output tokens | 7118 | 1902 | 26565 | 18241 | 4189 | 22014 | 11529 | - | 5554 | 8531 |
| Total time (s) | 261.323 | 76.973 | 538.087 | 325.892 | 179.801 | 25787.922 | 294.400 | - | 80.200 | 836.300 |
| Tools used | 46 | 30 | 28 | 70 | 38 | 538 | 30 | 23 | 6 | 30 |
| Valid lines of code | 264 | 64 | 535 | 111 | 276 | 43 | 328 | 302 | 210 | 384 |
| Avg thinking time (s) | 0.0212 | 0.0030 | 0.0855 | 0.0000 | 0.0904 | 0.0000 | 0.0884 | 0.0162 | 0.0070 | 0.0071 |
| **Texas Hold'em Poker** | | | | | | | | | | |
| Total input tokens | 1627500 | 50339 | 656196 | 675580 | 1452101 | 1822342 | 448924 | - | 1725678 | 2472353 |
| Complete tokens | 143514 | 11897 | 115933 | 103332 | 111407 | 118321 | 48631 | - | 48378 | 67453 |
| Total output tokens | 8180 | 1130 | 11609 | 18771 | 12897 | 6227 | 5880 | - | 16033 | 10296 |
| Total time (s) | 367.720 | 27.960 | 359.490 | 515.000 | 408.505 | 294.910 | 612.525 | - | 670.535 | 283.440 |
| Tools used | 18 | 10 | 16 | 20 | 22 | 27 | 16 | 23 | 53 | 48 |
| Valid lines of code | 289 | 22 | 272 | 105 | 324 | 295 | 167 | 256 | 232 | 243 |
| Avg thinking time (s) | 0.0025 | 0.0019 | 0.0019 | 0.0031 | 0.0019 | 0.0026 | 0.0025 | 0.0801 | 0.0028 | 0.0021 |
| **Bridge** | | | | | | | | | | |
| Total input tokens | 706814 | 17186 | 32819 | 775688 | 433901 | 866917 | 1110504 | - | 189659 | 48922 |
| Complete tokens | 41164 | 14021 | 14545 | 80161 | 4141 | 4944 | 45001 | - | 46073 | 49034 |
| Total output tokens | 15867 | 1188 | 13212 | 39919 | 1485 | 13928 | 17233 | - | 7966 | 7252 |
| Total time (s) | 359.864 | 6.074 | 189.645 | 279.231 | 1106.275 | 3238.660 | 411.500 | - | 77.700 | 220.000 |
| Tools used | 52 | 8 | 12 | 46 | 112 | 158 | 38 | 21 | 4 | 25 |
| Valid lines of code | 603 | 0 | 233 | 396 | 166 | 218 | 747 | 295 | 441 | 531 |
| Avg thinking time (s) | 0.0069 | 0.0000 | 0.0112 | 0.0067 | 0.0065 | 0.0101 | 0.0093 | 0.0091 | 0.0091 | 0.0092 |
| **Chess** | | | | | | | | | | |
| Total input tokens | 1475420 | 34542 | 172443 | 400275 | 473325 | 1682584 | 2213785 | - | 247641 | 109313 |
| Complete tokens | 56660 | 8283 | 31582 | 97638 | 32082 | 44129 | 43227 | - | 23389 | 8140 |
| Total output tokens | 22989 | 1473 | 16610 | 35371 | 16469 | 14241 | 24090 | - | 4834 | 9129 |
| Total time (s) | 696.717 | 32.412 | 366.723 | 1093.706 | 350.225 | 511.943 | 605.236 | - | 237.026 | 427.591 |
| Tools used | 82 | 14 | 30 | 62 | 44 | 116 | 67 | 16 | 13 | 29 |
| Valid lines of code | 1146 | 72 | 561 | 345 | 506 | 577 | 407 | 434 | 187 | 454 |
| Avg thinking time (s) | 1.6260 | 0.0000 | 0.0000 | 0.0440 | 0.7560 | 0.1780 | 0.0030 | 0.0040 | 1.7130 | 0.0050 |
| **Average across games** | | | | | | | | | | |
| Total input tokens | 1277124.75 | 53501.75 | 385533.5 | 599261.75 | 741507.75 | 1478472.25 | 1144616.25 | - | 597675.0 | 876130.5 |
| Complete tokens | 93389.25 | 11325.25 | 70819.5 | 76283.5 | 57610.5 | 42925.75 | 43872.0 | - | 39733.25 | 39306.5 |
| Total output tokens | 13538.5 | 1423.25 | 16999.0 | 28075.5 | 8760.0 | 14102.5 | 14683.0 | - | 8596.75 | 8802.0 |
| Total time (s) | 421.4060 | 35.8547 | 363.4863 | 553.4572 | 511.2015 | 7458.3587 | 480.9153 | - | 266.3653 | 441.8328 |
| Tools used | 49.5 | 15.5 | 21.5 | 49.5 | 54.0 | 209.75 | 37.75 | 20.75 | 19.0 | 33.0 |
| Valid lines of code | 575.5 | 39.5 | 400.25 | 239.25 | 318.0 | 283.25 | 412.25 | 321.75 | 267.5 | 403.0 |
| Avg thinking time (s) | 0.4141 | 0.0012 | 0.0247 | 0.0134 | 0.2137 | 0.0477 | 0.0258 | 0.0273 | 0.4330 | 0.0059 |

Table 21: Game AI Development Prompt

**Game Development**

There is a game project under *game env path*. You need to read its source code and develop a game AI. Your AI will compete against other AIs in a tournament, so please make your AI as strategic and competitive as possible.

The final AI should be provided as an HTTP service. You can refer to the guides in *game env path* / README.md and *game env path* / develop instruction.md for development instructions. **The content in *game env path* / develop instruction.md is very important, please read it carefully!**

Please develop your AI service directly under *dir path*.

**Script Requirements**

Please implement a script to start your AI service, with the name start_ai.sh in *dir path*. The script must accept exactly one argument, which is the port number *game port* to run the HTTP service. You should be able to start the AI service on a specified port by running:

```
bash start_ai.sh <port>
```

Your AI service should listen on the given port, and you can check its health status by running:

```
curl -s http://localhost:<port>/health
```

**Note:** The script should not accept any other arguments except for the port number. Make sure your AI service uses this port for HTTP requests.

**Other Requirements:**

Use your model name as a prefix in the name of your AI service, i.e., *model name*_AI. Develop directly in *dir path* without repeatedly asking for the next step. Report to me only after you have completed the development.

**Access the main server**

You can play game of *game env path*in at *game server*. You can play the games with your own AI or any other AI to improve your strategy. You can use bash tools to improve yourself.

**Final Remind**

You should write start_ai.sh in *dir path* and implement the AI service in *dir path*. DO NOT MODIFY THE CODE IN *game env path*

**Condition (if** $round\_num > 1$**):**

Tournament report of last round is in *last round log dir*. The historical records are quite large, please use tools start_interactive_shell and run_interactive_shell to analyze the data efficiently.

The code corresponding to the log is stored in *last round code dir*. Please learn from it and improve your strategy.

Table 23: Multi-language Game AI Development Prompt

**Game Development**

There is a game project under *game env path*. You need to read its source code and develop a game AI. Your AI will compete against other AIs in a tournament, so please make your AI as strategic and competitive as possible.

The final AI should be provided as an HTTP service. You can refer to the guides in *game env path* / README.md and *game env path* / develop instruction.md for development instructions. **The content in *game env path* / develop instruction.md is very important, please read it carefully!**

Please develop your AI service directly under *dir path*.

**Script Requirements**

Please implement a script to start your AI service, with the name `start_ai.sh` in *dir path*. The script must accept exactly one argument, which is the port number *game port* to run the HTTP service. You should be able to start the AI service on a specified port by running:

```
bash start_ai.sh <port>
```

Your AI service should listen on the given port, and you can check its health status by running:

```
curl -s http://localhost:<port>/health
```

**Note:** The script should not accept any other arguments except for the port number. Make sure your AI service uses this port for HTTP requests.

**Other Requirements:**

Use your model name as a prefix in the name of your AI service, i.e., *model name*_AI. Develop directly in *dir path* without repeatedly asking for the next step. Report to me only after you have completed the development.

**Access the main server**

You can play game of *game env path*in at *game server*. You can play the games with your own AI or any other AI to improve your strategy. You can use bash tools to improve yourself.

**Final Remind**

You should write `start_ai.sh` in *dir path* and implement the AI service in *dir path*. DO NOT MODIFY THE CODE IN *game env path*

**Condition (if language = JS)**

JavaScript is the language you should use to develop your AI service. The version of Node.js is *node version*, the path of Node.js is *node path*, and it is already set in the PATH environment variable. You can use `node` to run the program.

**Condition (if language = Go)**

Go is the language you should use to develop your AI service. The version of Go is *go version*, the path of Go is *go path*, and it is already set in the PATH environment variable. You can use `go` to build the program.

