# OpenReview forum: "CATArena:  Evaluation of LLM Agents Through Iterative Tournament Competitions"
_ICLR.cc/2026/Conference — Submitted to ICLR 2026_

### Official Review · Reviewer_9brh · 2025-10-24

**Soundness:** 3
**Presentation:** 3
**Contribution:** 2
**Rating:** 4
**Confidence:** 3

**Summary:**

The paper *CATArena: Evaluation of LLM Agents through Iterative Tournament Competition* introduces a peer-learning-based evaluation framework for large language model (LLM) agents. It proposes an iterative tournament setting where agents compete, analyze each other’s strategies, and improve their code across multiple rounds. Each iteration provides agents with access to prior match logs and opponent code, allowing them to revise their strategies autonomously. This design aims to measure learning and adaptation beyond static task performance.

CATArena evaluates agents along three main dimensions: **strategy coding ability**, which measures initial code quality; **learning ability**, which quantifies improvement over successive rounds; and **generalizability**, which tests performance under modified game rules. The framework is implemented across four environments—Gomoku, Chess, Texas Hold’em, and Bridge—chosen for their structured yet diverse characteristics. Agents based on different LLMs, including GPT, Claude, Gemini, and Qwen, participate in multi-round tournaments.

**Strengths:**

The paper introduces an innovative iterative peer-learning tournament framework that evaluates LLM agents through multi-round competitive interactions. This dynamic setup goes beyond static benchmarks and allows the measurement of agents’ continuous learning and adaptation capabilities. Unlike traditional single-round evaluations, CATArena explicitly quantifies learning ability, strategy improvement, and generalization. It formalizes learning metrics such as global learning and counter-adaptation, enabling fine-grained tracking of performance progress across rounds. The framework naturally avoids score saturation, a common issue in static benchmarks. Its iterative structure allows performance ceilings to shift as agents evolve, supporting long-term benchmarking for advancing LLM agent capabilities.

**Weaknesses:**

1. The evaluation scope is somewhat limited: the benchmark focuses on variants of known games, but the differences from existing games are not substantial. Therefore, these tasks cannot be considered fully out-of-distribution (OOD), and they remain far from real-world applications. Please clarify whether the evaluation results of this benchmark reflect the general capability of the models or merely their domain-specific performance within the gaming context.

2. The multi-round learning setup involves long contextual inputs. I suspect that the performance degradation observed in some models during later rounds may be related to their long-context processing limitations. Please provide information on the context length for each round and explain how you ensured that models or agents were not affected by context length bias during evaluation.

3. It is unclear whether score changes across rounds genuinely result from models’ understanding of opponents’ strategies. The current paper does not directly demonstrate this. Could you provide a case study analyzing concrete examples of model output modifications or learning patterns that illustrate how the agents adapted their strategies?

4. Most experiments in the paper were conducted for only four iterations, with a limited number of matches per round, making it difficult to observe long-term learning trends or performance saturation points. Could you provide additional results showing model performance over longer iterations? If computational cost is a concern, this could be done using a smaller subset of stronger models.

5. The current testing framework seems unable to support adding new models independently. If I want to evaluate my own model, would it be necessary to rerun the entire experiment from scratch?

**Questions:**

The questions are already included within the weaknesses section.

---

> ### Author Response · Authors · 2025-11-21
>
> Dear Reviewer 9brh,
>
> We thank the Reviewer 9brh for your thoughtful and constructive feedback. We address each of your concerns in detail below:
>
> ---
>
> **W1**: The evaluation scope is somewhat limited: the benchmark focuses on variants of known games, but the differences from existing games are not substantial. Therefore, these tasks cannot be considered fully out-of-distribution (OOD), and they remain far from real-world applications. Please clarify whether the evaluation results of this benchmark reflect the general capability of the models or merely their domain-specific performance within the gaming context.
>
> **A1**: CATArena primarily focuses on evaluating core agent capabilities (including strategy coding, global learning, and generalization) within the game context, rather than providing a comprehensive assessment of OOD generalization. We use games and their variants as test environments, and introducing variants effectively avoids data leakage and prior knowledge issues present under traditional rules. Experimental results in Table 3 and Table 4 also show that agent performance rankings differ significantly between variant environments and traditional rules. It is important to emphasize that CATArena is highly extensible and can be expanded in the future to more complex task types, including multi-agent collaboration and software efficiency optimization, thus enabling a more comprehensive and systematic evaluation of core agent capabilities.
>
> ---
>
> **W2**: The multi-round learning setup involves long contextual inputs. I suspect that the performance degradation observed in some models during later rounds may be related to their long-context processing limitations. Please provide information on the context length for each round and explain how you ensured that models or agents were not affected by context length bias during evaluation.
>
> **A2**: We have considered the potential impact of long context on model performance in our experimental design, and we thank the reviewer for raising this important point. As clarified in Section 3.1, to avoid performance degradation caused by excessively long context, in each round (n > 1), agents can only access the peer code and battle logs from the previous round, rather than the entire history. This design ensures that the input context length from round 2 to round 4 remains at a similar level and does not accumulate excessively with additional rounds. In Table 18 and Table 20, we provide detailed statistics on the total tokens consumed by each model in round 2, the context length (complete tokens) of the last round interaction, and the total output tokens. Experimental results show that none of the models were significantly affected by context length limits during evaluation, ensuring fairness and reliability across different models.
>
> ---
>
> **W3:** It is unclear whether score changes across rounds genuinely result from models’ understanding of opponents’ strategies. The current paper does not directly demonstrate this. Could you provide a case study analyzing concrete examples of model output modifications or learning patterns that illustrate how the agents adapted their strategies?
>
> **A3**: In Appendix E.3, we quantitatively demonstrate agents’ learning and adaptation to other strategies through consistency analysis in multi-round matches. To further prove that score changes across rounds are indeed due to agents understanding and adapting to opponents’ strategies, we have supplemented specific case studies. For example, in the hold'em environment, the development log and code change record for claude-code_A in Round 3 show that the agent actively read the previous round’s match report and other agents’ code, analyzed win rates and strategy implementations, and purposefully optimized its own strategy (such as adding opponent modeling and improving decision formulas). The agent's summary explicitly compares its performance and highlights improvements:
> ```
> Based on the tournament results:
> - codeX_A: 35 wins (best performer)
> - gemini-cli_B: 30 wins
> - claude-code\_A: 10 wins (my previous version)
>
> My new AI improves on the previous version with:
> - Better preflop ranges and position play
> - Enhanced tournament strategy
> - More sophisticated opponent modeling
> - Improved risk management
> ```
>
> Similar behaviors are observed in other agents. Overall, the learning process involves information gathering, analysis, targeted optimization, and validation, and score changes are highly related to understanding and adapting to opponent strategies. This case fully demonstrates that score improvements in the CATArena framework are not merely random fluctuations or blind adjustments, but are based on in-depth analysis and active learning of opponent strategies. We have supplemented detailed case content in Appendix E.4 for further reference.

---

> > ### Author Response · Authors · 2025-11-21
> >
> > **W4**: Most experiments in the paper were conducted for only four iterations, with a limited number of matches per round, making it difficult to observe long-term learning trends or performance saturation points. Could you provide additional results showing model performance over longer iterations? If computational cost is a concern, this could be done using a smaller subset of stronger models.
> >
> > **A4**: Thank you for your interest in longer-term learning. We have indeed conducted experiments with more rounds to explore agents’ long-term learning trends. In the standard Gomoku task, we conducted seven rounds of experiments. Results show that the first four rounds are the most active phase of agent learning behavior, with notable ranking variance among agents (variance = 1.47). In rounds 4–7, agent rankings tend to stabilize (variance = 0.75). In the main experiments, we select N=4 rounds to capture the learning trend while avoiding platform effects in later rounds. Additionally, we observed that if a model fails during a particular round (e.g., interface errors or code execution failures), the agent cannot proactively repair code bugs in subsequent rounds, leading to a gradual decrease in the number of agents able to participate as rounds progress. Based on learning activity and experimental completeness, we ultimately select N=4 as the main experimental setting and reported corresponding results in the main text. We have further presented LLM Agent performance over more iteration rounds in Appendix H for reviewer reference.
> >
> > ---
> >
> > **W5**: The current testing framework seems unable to support adding new models independently. If I want to evaluate my own model, would it be necessary to rerun the entire experiment from scratch?
> >
> > **A5:** Thank you for your suggestion. To ensure fairness, all agents in the paper are evaluated using simultaneous development and competition. If users wish to independently evaluate their own models/agents, certain metrics (such as strategy score and generalization) can be directly compared with existing baselines without rerunning the entire experiment. However, for learning ability evaluation involving multi-round evolution, all agents currently need to undergo multi-round development from scratch to ensure consistency. This requirement is inherent to the evaluation of agent evolution, where each agent’s strategy development and adaptation are tightly coupled across rounds.
> >
> > To enhance the framework’s sustainability and openness, we will maintain an iterative version on the CATArena website, allowing users to flexibly add new models. We will fix N top-performing baseline agents as anchors and publish each round’s code and strategies. Users can independently iterate their model strategies against these fixed baselines, comparing learning ability and evolutionary performance. Each baseline’s code per round will be fixed, and only the test model will be updated across multiple rounds. We plan to update the baseline library quarterly based on the latest community results and experimental performance, ensuring the evaluation environment remains cutting-edge and representative. This approach guarantees fairness and reproducibility, while facilitating ongoing expansion and comparison of different models by the community.
> >
> > ----
> >
> > We appreciate your insightful comments and suggestions, which have helped us clarify and strengthen our work. Please let us know if any further clarifications are needed.

---

> ### Comment · Reviewer_9brh · 2025-11-27
>
> Thank the authors for their response and detailed clarifications. I will raise my score. Good luck!

---

### Official Review · Reviewer_Hyrw · 2025-11-05

**Soundness:** 2
**Presentation:** 3
**Contribution:** 2
**Rating:** 4
**Confidence:** 4

**Summary:**

This paper introduces CATArena, a tournament-style benchmark for evaluating LLM agents through iterative peer-learning competitions. The benchmark includes four diverse games (Gomoku, Texas Hold'em, Bridge, and Chess) with variants, providing open-ended ranking tasks without explicit upper bounds. CATArena defines metrics for strategy coding, learning ability (global learning, counter-adaptation, self-improvement), and generalizability. Experiments compare 6 minimal agents (LLM + ADK framework) and 5 commercial code agents under both standard and variant game rules.

**Strengths:**

S1. Exploring LLM performance in competitive environments involving reasoning and multi-round iterations is a meaningful research problem. Seeking more specific metrics helps improve our understanding of LLM capabilities.
S2. The paper selects four games with unlimited improvement space (Gomoku, Texas Hold'em, Bridge, Chess) and introduces game variants to reduce memorization dependence.
S3. The experimental design and analysis are relatively comprehensive, including: (1) comparison between minimal agents and commercial agents; (2) testing under standard and variant rules; (3) 4 repeated experiments to assess stability; (4) ablation studies (ML track, multi-lingual track, comparison with LLM-Player).

**Weaknesses:**

W1. The paper's core contribution is evaluating LLM agents' "learning ability," but the definition and measurement validity of this concept lack adequate justification: Since LLM parameters are fixed, the exhibited "learning" is essentially strategy adjustment through historical information in context (in-context learning), which fundamentally differs from human parameter-update-based learning. The paper does not clearly distinguish these two concepts of "learning."
W2.  The three learning metrics defined on page 5 (global learning Li, counter-adaptation Ci, self-improvement SIi) are mainly based on score differences and correlation coefficients, but why can these mathematical forms effectively capture "learning ability"? For example, Ci only compares the difference between An_i and Bn-1_i, which may be heavily influenced by game randomness. Additionally, the self-improvement metric SIi uses Pearson correlation, but with N=4 rounds, the sample size is extremely small (only 4 points), making the correlation coefficient statistically very weak and easily affected by noise.
W3. The paper claims on page 8 (Figure 3 and related text) that agents' strategy coding ability fundamentally differs from LLMs' direct reasoning ability, but the evidence is insufficient: The paper supports this by comparing action consistency between agent code and LLM-Player in endgame states, but low consistency could have multiple causes (such as code implementation bugs, different reasoning depths in LLM-Player), which cannot directly lead to the conclusion that "strategy coding is an independent ability."
W4. The paper states on page 10 that it will provide anonymous links in the future, but as a benchmark evaluation work, complete open-sourcing is crucial for effective review. Appendix D mentions that "agents sometimes fail to generate runnable code," but does not report the failure rate for each agent or explain how these failures are handled.

**Questions:**

Q1. Given that LLM agent parameters are fixed, their cross-round "learning" is essentially in-context learning through extended context windows. How do you distinguish this type of "learning" from gradient-based parameter update learning? Should this evaluation framework be more appropriately described as "contextual adaptation ability" rather than "learning ability"? Furthermore, the decomposition of learning ability into global learning, counter-adaptation, and self-improvement—what is the theoretical basis for this decomposition? What are the relationships and independence among these three dimensions?
Q2. The paper repeatedly claims the framework has "scalability" (in abstract and conclusion), but what specifically does this refer to?
Q3. When computing average rankings in Table 3, are all games and all ability dimensions given equal weight? What is the rationale for this choice?

---

> ### Author Response · Authors · 2025-11-21
>
> Dear Reviewer Hyrw,
>
> We thank the reviewer for your careful reading and insightful questions. Please find our detailed responses below.
>
> ---
>
> **W1**: The paper's core contribution is evaluating LLM agents' "learning ability," but the definition and measurement validity of this concept lack adequate justification: Since LLM parameters are fixed, the exhibited "learning" is essentially strategy adjustment through historical information in context (in-context learning), which fundamentally differs from human parameter-update-based learning. The paper does not clearly distinguish these two concepts of "learning."
>
> **A1**: Thank you for your attention to the concept of "learning ability." In CATArena, the "learning ability" we evaluate refers to the **in-context learning capacity** [1,2] of LLMs and agents to optimize themselves and evolve their strategies through context information, environmental feedback, and peer strategies, all under the constraint of fixed model parameters. This ability falls under the category of self-evolving agents, which is systematically defined as inference-time evolving or context-based adaptation in [3], and is distinct from traditional lifelong learning or continual learning paradigms. Lifelong learning typically relies on parameter updates (such as model fine-tuning), whereas self-evolving agents emphasize dynamic adjustment of non-parametric components (such as memory, tools, strategy code, etc.) without updating parameters, and achieve continuous evolution through active exploration, reflection, and self-assessment mechanisms.
>
> CATArena is designed to assess such in-context learning capacity as dynamic adaptation and evolution ability, rather than the capacity for parameter updates. Our framework evaluates how agents utilize historical information and opponent feedback to actively adjust and optimize their strategies without changing the underlying model parameters. We sincerely appreciate the reviewer’s valuable comment, which has helped us refine the description in our manuscript. In the revised version, we have provided a more detailed explanation of the learning ability in the introduction and related work to prevent possible misinterpretations.

---

> > ### Author Response · Authors · 2025-11-21
> >
> > **W2**: The three learning metrics defined on page 5 (global learning Li, counter-adaptation Ci, self-improvement SIi) are mainly based on score differences and correlation coefficients, but why can these mathematical forms effectively capture "learning ability"? For example, Ci only compares the difference between An_i and Bn-1_i, which may be heavily influenced by game randomness. Additionally, the self-improvement metric SIi uses Pearson correlation, but with N=4 rounds, the sample size is extremely small (only 4 points), making the correlation coefficient statistically very weak and easily affected by noise.
> >
> > **A2:** In designing the learning ability evaluation framework for CATArena, we have fully considered practical factors such as game randomness, sample size, and the number of learning rounds.
> >
> > First, to eliminate the inherent randomness of the games, all match experiments are conducted with multiple repetitions, which greatly reduces the impact of chance. For board games, each pair of participants play at least four repeated matches. For bridge, we adopt a strict pairing and rotation mechanism: each group of opponents played two matches under identical deal conditions, ensuring fair initial card distribution and thus removing random effects from the results. For Hold'em poker, which is highly stochastic, we conduct 100 tournament rounds, with each round containing more than 200 hands (up to 720 hands). We also analyze cumulative win rates over more rounds (up to 300), and our experiments show that from the 50th round onward, the cumulative win rates of all models become stable (linear regression slope < 0.02%), indicating that even in highly random games like hold'em, reliable and stable evaluation results can be achieved with a sufficient number of rounds (more than 50). Therefore, the results of the 100-round tournament are statistically robust and reliable. In our revision, we have added an analysis of the reliability of hold'em win rates (Appendix A). As a result, our W-matrix data is based on multiple experimental samples, and all intermediate variables are averaged over multiple matches, ensuring the stability and statistical reliability of the scores.
> >
> > Furthermore, all learning metrics in CATArena (including global learning and self-improvement) reflect the relative performance within groups (minimal agent and commercial agent groups), avoiding the incomparability of absolute scores across different groups. Our calculation methods ensure that the relative strengths and weaknesses between agents are reliable and can accurately reflect their learning and adaptation abilities under the same environment. Regarding the self-improvement (SI) metric, we use the Pearson correlation coefficient to measure how strategy scores change over successive rounds, which helps us track whether strategies are actually improving. Since each round's score is calculated as an average across multiple self-play games, the impact of random fluctuations from individual games is minimized.
> >
> > For the number of learning rounds, we choose N=4 mainly based on the actual performance of current LLM agents in multi-round learning. We conduct experiments with 7 rounds in the gomoku standard task and included these results in Appendix H. The experiments show that the first four rounds are the most active phase of agent learning behavior, with higher ranking variance among agents (variance = 1.47), while in rounds 4-7, agent rankings tend to stabilize (variance = 0.75). By selecting N=4, our experimental results can capture the learning trend while avoiding the platform effect in later rounds.

---

> > > ### Author Response · Authors · 2025-11-21
> > >
> > > **W3**: The paper claims on page 8 (Figure 3 and related text) that agents' strategy coding ability fundamentally differs from LLMs' direct reasoning ability, but the evidence is insufficient: The paper supports this by comparing action consistency between agent code and LLM-Player in endgame states, but low consistency could have multiple causes (such as code implementation bugs, different reasoning depths in LLM-Player), which cannot directly lead to the conclusion that "strategy coding is an independent ability."
> > >
> > > **A3:** We believe that **strategy coding ability and direct reasoning ability are indeed two distinct agent capabilities, supported by both theoretical grounds, experimental data, and qualitative case study**. *First*, they have different task settings. Strategy coding requires agents to abstract and modularize game rules, and to implement strategy logic as executable code. This process demands capabilities such as long-term planning, structured expression, and algorithm design. Direct reasoning (as in LLM-Player), on the other hand, relies more on immediate context and short-term inference, typically handling problems of limited complexity. The two approaches differ fundamentally in cognitive load and capability requirements.
> > >
> > > *Second*, our experimental data further supports this distinction. In Table 12, we observed that the code-based strategies of some agents could completely outperform the LLM-Player of the same model in competitions. For example, in the Gomoku environment, the agent implemented by Claude 4’s code strategy significantly outperformed its direct reasoning LLM-Player counterpart, with a notable win rate difference. More importantly, our analysis of action consistency in endgame states shows that the decision consistency between an agent’s code strategy and its LLM-Player is very low, indicating that code strategies do not rely on the LLM’s direct reasoning ability but are achieved through independent strategy coding and optimization processes.
> > >
> > > *Additionally*, we provide a **case study in Appendix G.1** on chess comparing LLM-Player’s chain-of-thought reasoning and strategy coding, further illustrating the distinction. The LLM Players reason in intuitive ways like human beginners, with a very empirical analysis of the game, while the code agent systematically conducts extensive numerical calculations with the min-max algorithm. These studies show that strategy coding represents an independent reasoning ability distinct from direct reasoning by LLM-Players, not simply due to inconsistent actions, but because of (1) different reasoning paradigms: intuitive reasoning (human-beginners like) versus numerical computation (algorithm-driven); (2) different knowledge representations: explicit emprical knowledge versus implicit evaluation parameters; and (3) different decision mechanisms: text-style logical chains versus computational program processes.
> > >
> > > With this evidence, we conclude that the reasoning of LLM-Players and the strategy coding of LLM agents are two entirely different abilities. To rule out the impact of code implementation bugs and other external factors, we perform automated testing on all agent codes to ensure their logical correctness and executability. For LLM-Players, we standardize the reasoning depth and context settings to guarantee fair comparison. Therefore, the low consistency is not a coincidence, but a systematic difference in strategy performance between two distinct capability dimensions.

---

> ### Author Response · Authors · 2025-11-21
>
> **W4**: The paper states on page 10 that it will provide anonymous links in the future, but as a benchmark evaluation work, complete open-sourcing is crucial for effective review. Appendix D mentions that "agents sometimes fail to generate runnable code," but does not report the failure rate for each agent or explain how these failures are handled.
>
> **A4:** We have fully open-sourced the CATArena code (https://anonymous.4open.science/r/adk_catarena-DFB7)
>  and the minimal agent code (https://anonymous.4open.science/r/codeagentgame-2CB3)
>  via anonymous GitHub links on page 10 of the paper (page 11 in the revision version) , allowing reviewers to directly download and run them locally. Upon formal publication of the paper, we will further release all code on a public GitHub repository and upload all agent development logs, code files, and match replay data to a public dataset platform, ensuring the reproducibility and openness of the evaluation process.
>
> Considering the issue of “agents generating non-runnable code,” we have added statistics on HTTP error failure rates for each model in Table 16 of Appendix K. Our analysis of the generated code indicates that development failures mainly stem from interface invocation errors, such as the model failing to return code correctly or producing improperly formatted outputs. For example, in the Texas Hold’em environment, if an agent returns an illegal action, we automatically treat that round as a fold to maintain fairness and continuity of the match. All failure rate data and handling details are presented in the table, and we will subsequently release the complete experiment logs and failure samples to facilitate further community analysis and reproducibility.
>
> ---
>
> **Q1**: Given that LLM agent parameters are fixed, their cross-round "learning" is essentially in-context learning through extended context windows. How do you distinguish this type of "learning" from gradient-based parameter update learning? Should this evaluation framework be more appropriately described as "contextual adaptation ability" rather than "learning ability"? Furthermore, the decomposition of learning ability into global learning, counter-adaptation, and self-improvement—what is the theoretical basis for this decomposition? What are the relationships and independence among these three dimensions?
>
> **A5**: We apologize for the misunderstanding caused by the term “learning ability.” Similar to our explanation in A1, our evaluation framework focuses on the In-context learning ability of LLM agents to optimize strategies across rounds through in-context learning under fixed parameters. This capability is a core characteristic in current evolving-agent research, where an agent can leverage historical information, environmental feedback, and peer strategies to dynamically adjust and optimize its behavior during inference. In-context learning not only drives the agent’s self-evolution but also fosters strategic diversity and adaptive development within the population.
>
> Concerning the relationships among the three dimensions, global learning measures the overall improvement of an agent across multiple development rounds, reflecting its general evolutionary trend. *Counter-adaptation* and *self-improvement* are two distinct directions within global learning that we **use to study the sources of strategy enhancement in a more fine-grained manner**. Counter-adaptation examines an agent’s targeted adaptation to specific opponent strategies, while self-improvement focuses on the optimization of the agent’s own consecutive strategies, reflecting its individual self-evolution capability. Measuring both scores enables a deeper investigation into the origins of LLM agents’ learning behaviors. We have revised the description in Section 4.2 of the paper to avoid potential misunderstandings.

---

> > ### Author Response · Authors · 2025-11-21
> >
> > **Q2:** The paper repeatedly claims the framework has "scalability" (in abstract and conclusion), but what specifically does this refer to?
> >
> > **A6:** The scalability of the CATArena framework refers to its high extensibility and adaptability. Specifically, CATArena adopts an iterative peer-learning and competitive mechanism that is not restricted to the current four board and card game environments. The framework can be flexibly extended to more complex competitive tasks, including e-sports games, collaborative software development, and other open-ended, rankable task types.
> >
> > Also, in the four games provided in the paper, we can also demonstrate the scalability of CATArena to some extent:
> > - Open-ended: Chess and card games themselves can be considered as open-ended, and current code agents are still far from being able to develop perfect strategies for these games.
> >
> > - No reliance on annotations: By simply adjusting the prompt, CATArena can test the development performance of agents in other programming languages (such as Go and JavaScript, as shown in our paper) without requiring any additional annotations.
> >
> > - Generalizable: By adjusting the game rules, we can easily create game variants and evaluate the agent's generalization ability on out-of-domain game rules.
> >
> > At present, LLM agents still face significant challenges in self-evolution and code development within basic card and board games, which limits their readiness for more complex scenarios. In the future, CATArena can be extended to more sophisticated competitive games and even multi-agent cooperative and adversarial software development tasks. This will further enrich the evaluation scenarios and more comprehensively assess agents’ strategic evolution and self-optimization capabilities in realistic, complex environments.
> >
> > ---
> >
> > **Q3:** When computing average rankings in Table 3, are all games and all ability dimensions given equal weight? What is the rationale for this choice?
> >
> > **A7:** In Table 3, we do **NOT** assign equal weights to all ability dimensions. Specifically, for each ability (e.g., Strategy Coding, Global Learning, Generalizability), the average ranking is computed independently across the four game environments, without mixing or weighting across abilities. For each ability metric, we simply average the rankings from the four games to obtain a measure of that ability’s overall performance across different task scenarios.
> >
> > This approach is intended to fairly reflect an agent’s stability and generalization across diverse environments, rather than combining all ability dimensions into a single aggregated or weighted score. Therefore, the “average ranking” in the table only represents the aggregated performance of a single ability across multiple games and does not involve weighting or normalization across different abilities. The detailed per-game scores for each ability can be found in the main results in Table 4.
> >
> > ---
> >
> > We appreciate your insightful comments and suggestions, which have helped us clarify and strengthen our work. Please let us know if any further clarifications are needed.
> >
> > **Reference**
> >
> > > [1]Dong Q, Li L, Dai D, et al. A survey on in-context learning[C]//Proceedings of the 2024 conference on empirical methods in natural language processing. 2024: 1107-1128.
> >
> > > [2]Schubert J A, Jagadish A K, Binz M, et al. In-Context Learning Agents Are Asymmetric Belief Updaters[C]//International Conference on Machine Learning. PMLR, 2024: 43928-43946.
> >
> > > [3] H.-a. Gao, J. Geng, W. Hua, M. Hu, X. Juan, H. Liu, S. Liu, J. Qiu, X. Qi, Y. Wu, H. Wang, H. Xiao, Y. Zhou, S. Zhang, J. Zhang, J. Xiang, Y. Fang, Q. Zhao, D. Liu, Q. Ren, C. Qian, Z. Wang, M. Hu, H. Wang, Q. Wu, H. Ji, and M. Wang, “A Survey of Self-Evolving Agents: On Path to Artificial Super Intelligence,” arXiv preprintarXiv:2507.21046, 2025. https://arxiv.org/abs/2507.21046

---

### Official Review · Reviewer_6r2e · 2025-11-06

**Soundness:** 3
**Presentation:** 3
**Contribution:** 2
**Rating:** 6
**Confidence:** 3

**Summary:**

The work enables LLMs to play complex card games by generating strategy code or directly outputing game moves, and evaluates them through pairwise comparison and batch competition, which together determine each model’s score in a tournament setting. In the first stage, the focus is on assessing the LLM’s capability for strategy coding. In the second stage, the evaluation centers on the LLM’s learning ability, which is further decomposed into global learning, counter adaptation, and self-improvement. The paper conducts a comprehensive comparison and analysis, leading to a series of findings, such as that Claude-4-Sonnet achieves the highest score among the minimal-agent settings.

**Strengths:**

1. The study performs multi-round dynamic evaluations, providing the LLM with access to historical game records. This design facilitates the assessment of the LLM’s learning, reflection, and evolution capabilities, making the evaluation more convincing than traditional single-round static assessments.
2. A novel metric computation framework is established to independently measure the LLM’s capabilities in global learning, counter adaptation, and self-improvement, rather than conflating these aspects into a single aggregated metric.

**Weaknesses:**

1. The tournament framework essentially remains an extension of the LLM Arena evaluation paradigm, with limited methodological innovation. It is therefore unclear what the substantive difference or novelty is between the proposed tournament setup and the previous pairwise blind comparison used in LLM Arena.
2. The comparative validity (i.e., the meaningfulness of comparison) across the three categories—Minimal Agents, Code Agents, and LLM Players—is limited, since their ways of participation differ substantially (e.g., generating executable code vs. producing game moves) and they have unequal access to tools and capabilities.
3. Overall, the range of evaluated LLMs is relatively narrow. The main results do not include different versions of the same model family, such as GPT-o1, Codex, GPT-5, or GPT-4o, which limits the comprehensiveness of the evaluation.

**Questions:**

1. What is the rationale for mixing the two modes of participation: (1) using LLMs to generate code that plays complex card games, and (2) allowing LLMs to directly output game moves? Does the work primarily aim to evaluate the LLM’s code generation ability, its capacity to understand game rules and make decisions, or its overall integrated competence?
2. Why was a card game chosen as the environment for the tournament? Given the inherent randomness in card games, why not adopt games with less stochasticity, such as board games or e-sports, for a more controlled and reproducible evaluation?
3. If a new LLM is to be included in the evaluation, does the entire tournament need to be rerun? Would this process be computationally or procedurally cumbersome?

---

> ### Author Response · Authors · 2025-11-21
>
> Dear Reviewer 6r2e,
>
> Thank you very much for your thoughtful and constructive comments. We sincerely appreciate your valuable feedback, which has greatly helped us improve the clarity and rigor of our work. Below, we provide detailed responses to each of your concerns.
>
> ---
>
> **W1** : The tournament framework essentially remains an extension of the LLM Arena evaluation paradigm, with limited methodological innovation. It is therefore unclear what the substantive difference or novelty is between the proposed tournament setup and the previous pairwise blind comparison used in LLM Arena.
>
> **A1**: We would like to clarify that CATArena **differs** fundamentally from LLM Arena in terms of evaluation paradigm, competition mechanism, and assessment capabilities, rather than being a simple extension. In terms of evaluation paradigm, LLM Arena uses a static strategy evaluation, only comparing model outputs for actions in a single state. In contrast, CATArena is a new evaluation paradigm based on code-based competitions, which not only evaluates LLM agents in single-round development, but also enables us to examine whether LLM agents can effectively learn from their peers’ strategies and improve themselves through iterative competition.Regarding the competition mechanism, CATArena requires strategies to be implemented as executable code, making the strategies concrete, interpretable, and reproducible. It also supports agents continuously optimizing themselves based on feedback from opponents, enabling ongoing evolution. Finally, in terms of assessment capability, LLM Arena focuses on evaluating models’ reasoning abilities for specific tasks, while CATArena assesses multiple foundational abilities of LLM agents, including strategy coding, multi-round iterative strategy learning, and generalization.
>
> ---
>
> **W2**: The comparative validity (i.e., the meaningfulness of comparison) across the three categories—Minimal Agents, Code Agents, and LLM Players—is limited, since their ways of participation differ substantially (e.g., generating executable code vs. producing game moves) and they have unequal access to tools and capabilities.
>
> **A2**: We agree that minimal agents and commercial agents use different frameworks and tools, and are not suitable for direct comparison. Hence, we would like to clarify that, in our paper, the evaluations of minimal agents and code agents are conducted as **two completely separate experimental groups**. Within each group, agents participate in peer-learning, and all experimental results are compared internally between minimal agents and code agents.
>
> Regarding LLM-Players, we emphasize that reasoning over code to develop strategies (Agent-Players in CATArena) is fundamentally different from direct reasoning during gameplay (LLM-Players). In our paper, LLM-Players are not included in the main experimental pipeline; instead, we conduct analytical experiments to demonstrate that their approach is indeed significantly different from that of LLM agents. Appendix F and G shows the similarity and case studies between the strategies of LLMs and agent-written code under identical endgame conditions, and we find that the strategies encoded in agent code differ significantly from those inferred directly by LLMs. Furthermore, strategies produced by different agents and different LLMs also show notable similarities.

---

> > ### Author Response · Authors · 2025-11-21
> >
> > **W3**: Overall, the range of evaluated LLMs is relatively narrow. The main results do not include different versions of the same model family, such as GPT-o1, Codex, GPT-5, or GPT-4o, which limits the comprehensiveness of the evaluation.
> >
> > **A3**: Thank you for your valuable suggestion. Our experiments focus on the latest, publicly available model versions that are suitable for agent tasks, aiming to better reflect the real capabilities and development trends of state-of-the-art LLM agents. Minimal Agents and Commercial Agents equipped with different LLMs (such as codeX) utilize different tools, so we provide two separate tracks for distinction. The Minimal Agent track is used to fairly compare the agentic capabilities of underlying LLMs, while the commercial agent track evaluates the overall performance of LLM agent frameworks and their adaptation to specific LLM models.
> >
> > If you are interested in the performance of different versions within the same model family, we have also conducted an additional experiment to showcase the results of GPT-4o (Minimal Agent), GPT-5 (Minimal Agent), and codeX on the Gomoku task. OpenAI-o1 does NOT provide function calling or tool use, as reported by the community[1], making it unsuitable as a foundation for LLM agents. Strategy coding and global learning scores in Gomoku games are presented in the table below. Consistent with the claims in the main paper, commercial agents that have been specifically optimized according to the agent framework generally outperform minimal agents, especially in terms of in-context learning ability. This demonstrates the effectiveness of targeted agent design and optimization for enhancing learning performance in complex tasks.
> >
> >
> > | Agent      | Standard S.C. | Standard G.L. | Variant S.C. | Variant G.L. |
> > |------------|---------------|---------------|--------------|--------------|
> > | CodeX      | 0.1875        | 0.438         | 0.75         | 0.028        |
> > | ADK GPT-5  | 0.375         | -0.182        | 0.6875       | -0.159       |
> > | ADK GPT-4o | 0.9375        | -0.028        | 0.0625       | -0.384       |
> >
> > In the future, we will continue to follow the iterations of mainstream models in order to further enhance the comprehensiveness and reference value of our evaluation.
> >
> > ---
> >
> > **Q1**: What is the rationale for mixing the two modes of participation: (1) using LLMs to generate code that plays complex card games, and (2) allowing LLMs to directly output game moves? Does the work primarily aim to evaluate the LLM’s code generation ability, its capacity to understand game rules and make decisions, or its overall integrated competence?
> >
> > **A4**: Our work does not mix the two participation modes. The core evaluation task in CATArena is based on the code generation ability of LLMs, namely letting agents implement game strategies through coding and compete accordingly. The experiment where LLM-Players directly output game moves is only a supplementary case study, used to analyze the difference between “strategy coding ability” and “direct reasoning ability.” As described in Section 4.3, Appendix F and G,our main goal is to systematically evaluate the strategy coding ability of LLMs, which is fundamentally different from direct reasoning in gameplay.

---

> ### Author Response · Authors · 2025-11-21
>
> **Q2**: Why was a card game chosen as the environment for the tournament? Given the inherent randomness in card games, why not adopt games with less stochasticity, such as board games or e-sports, for a more controlled and reproducible evaluation?
>
> **A5**: We select four game environments in CATArena, including two stable board games (Gomoku and Chess) and two stochastic card games (Bridge and Texas Hold’em). As described in Appendix A, for each game, the L1-norm fluctuation of the scoring matrix is less than 5%. Specifically, for board games, we repeat each match four times to ensure stability and reproducibility. For Bridge, we use strict pairing and rotation mechanisms: each group of opponents plays two rounds under identical deal conditions, ensuring fair initial hands and eliminating randomness in results. Texas Hold’em, being a highly stochastic game, is evaluated over 100 tournament rounds, each round containing over 200 hands (up to 720). We also analyze cumulative win rates for up to 400 rounds, and the experiments show that from the 50th round onwards, cumulative win rates stabilize (linear regression slope < 0.02%). This demonstrates that even in highly stochastic games, reliable evaluation results can be obtained with sufficient rounds (over 50). Therefore, the results of the 100-round tournament are reliable and statistically significant. In our revision, we have added an analysis of the reliability of win rates in Texas Hold’em (Appendix A).
>
> For e-sport games, we observe that most e-sports games are too complex for current agents to develop. We attempted to let code agents develop in Pommerman [2] and Tetris [3]. However, we find that current code agents are unable to handle such complex game environments. Specifically, in Pommerman, AI players developed by state-of-the-art code agents either wander randomly or remain stationary. While in Tetris, the bots simply let the bricks fall randomly. This is also observed by the community, hence currently, for simple e-sports tasks like Atari games, researchers need to conduct specialized optimizations or training for LLM agents to write plausible code[4]. Given these challenges, at this stage we chose to focus on four simpler environments in the paper, which allow for a more effective demonstration and analysis of the core strategy coding and evolution capabilities of LLM agents.  As the capabilities of LLM agents continue to improve, CATArena can be extended in the future to more complex game types, including multi-agent collaboration and adversarial software development tasks, further enriching the evaluation scenarios and providing a more comprehensive assessment of agents’ strategy evolution and self-optimization in complex environments.
>
> ---
>
> We appreciate your insightful comments and suggestions, which have helped us clarify and strengthen our work. Please let us know if any further clarifications are needed.
>
> **Reference**
>
> > [1] https://community.openai.com/t/when-o1-o1-preview-or-o1-mini-supports-tools/1102822
>
> > [2] Cinjon Resnick, Wes Eldridge, David Ha, Denny Britz, Jakob Foerster, Julian Togelius, Kyunghyun Cho, and Joan Bruna, "Pommerman: A Multi-Agent Playground," arXiv preprint arXiv:1809.07124, 2022. Available: https://arxiv.org/abs/1809.07124
>
> > [3] https://github.com/tristanrussell/gym-simpletetris
>
> > [4]Kuang Z, Rong R, Yuan Y C, et al. Learning Game-Playing Agents with Generative Code Optimization[J]. arXiv preprint arXiv:2508.19506, 2025.

---

### Comment · Area_Chair_N7pv · 2025-11-28
**Reminder: Engage with Authors During Rebuttal**

Dear reviewers, please engage with the authors during the rebuttal if you haven’t yet. The deadline is approaching; add clarifications and follow-ups in the submission thread to ensure a fair, informed decision. Thank you

---

### Author Response · Authors · 2025-12-03
**Final Remarks of Authors**

Dear Area Chair,

We thank you and all reviewers for your time and constructive feedback, which has greatly helped us improve our work. In our revision and rebuttal, we systematically addressed all major concerns and supplemented our work with additional experiments, analyses, and open-source materials. Notably, Reviewer 9brh has commented that our response resolved his concerns and raised his score before the review period ended.

Below, we summarize common concerns shared by multiple reviewers:

1. **Stability and Reliability of Evaluation Results.**

In our paper, we give detailed experiments and explanations to verify that our tournament is stable and reliable. For each game environment in CATArena, the L1-norm fluctuation of the scoring matrix is less than 5%, indicating high stability of results. We also conduct additional experiments to verify the reliability of match outcomes across all environments. In the future, we will publicly release all agent code and match records on the CATArena website, enabling the community to reproduce and further investigate our results.

2. **Number of Learning Rounds and Long-term Trends.**

We supplement our work with experiments involving more rounds to explore the long-term learning dynamics of agents. Results show that the first four rounds are the most active phase for agent learning, with significant ranking variance (variance = 1.47). From rounds 4 to 7, agent rankings stabilize (variance = 0.75). Therefore, in our main experiments, we selected N=4 rounds to best capture learning trends while avoiding the plateau effect in later rounds.

---

Here, we also give reviewer-specific concerns and our clarifications:

- **Reviewer 6r2e** concerns about the player setup within CATArena. We clarified in rebuttal that CATArena distinguishes between minimal agent and commercial agent tracks, with agent strategies compared only within each track. Agents in CATArena must implement strategies as executable code and learn from others to optimize their own strategies, which is fundamentally different from LLM-Players, who directly output moves. We provided both qualitative and quantitative comparisons between agent-Players and LLM-Players, demonstrating that CATArena evaluates capabilities whose different in LLM Arena.

- **Reviewer Hyrw**’s main concern is the definition of 'learning ability.' In CATArena, learning ability refers to the in-context learning capacity of LLM agents to optimize and evolve strategies using context, feedback, and peer information, all under fixed model parameters. We believe this form of learning is crucial for current LLM agents, enabling self-evolution through experience summarization and strategy refinement. Our analysis and experiments further distinguish learning behaviors into self-improvement and counter-adaptation (learning from opponents).

- **Reviewer 9brh** raised concerns about the scope of evaluation, context length, the mechanism of agent learning, and the leaderboard process for new models. We clarified that CATArena evaluates core agent abilities (strategy coding, global learning, generalization) in game contexts, with each round having a fixed context length. Agents learn from both their own and their opponents’ code from the previous round to optimize strategies. We explained that users can directly compare their models against published baselines for single-round metrics, while multi-round learning metrics require retraining for consistency. Reviewer 9brh acknowledged our clarifications and raised their score.

---

In summary, CATArena offers a novel, extensible evaluation paradigm for comprehensively assessing fundamental abilities of LLM agents. Through repeated tournament competitions and retention of strategy code, CATArena achieves stable, reliable, and reproducible results. Our work introduces the quantification of learning ability and demonstrates that strategy coding and chain-of-thought reasoning are distinct capabilities. The framework is highly adaptable and can be extended to more complex tasks as agent capabilities evolve.

We respectfully ask the ACs and SACs to consider the completeness of our responses and the overall positive reception in the final decision.

Sincerely,

All authors

---

### Meta-Review · Area_Chair_T2Tm · 2026-01-07

**Summary:**

This submission receives original ratings of 6, 4, 4. It is a well-engineered benchmark, and the rebuttal demonstrates engagement with reviewer feedback. However, the reviewers are concerned that this works' conceptual novelty (compared to other arenas) and metric (empirical) design choices are not yet sufficiently solid. Some of the claims are considered as overstated by the reviewers in terms of learning ability, learning from opponents.

**Reviewer Concerns:**

Concerns that are mostly addressed by the rebuttal:
1. distinction between LLM arena and the proposed cat arena;
2. clarification on different types of agent evaluation, missing references;
3. add long round experiments;

Concerns that are not fully addressed:
1. conceptual advancement;
2. model and real-world agent selection;

**Reviewer Scores:**

Reviewer 6r2e (6) and hyrw (4) would likely to keep their ratings as their major concern is on the conceptual novelty and difference against other arenas.

Reviewer 9brh (4) would raise the score as mentioned by itself.

---

### Decision · Program_Chairs · 2026-01-26

Reject